

# Relativistic reduced density matrix functional theory

**Mauricio Rodríguez-Mayorga⋆, Klaas J. H. Giesbertz and Lucas Visscher**

Department of Theoretical Chemistry, Vrije Universiteit Amsterdam,
De Boelelaan 1083, 1081 HV Amsterdam, The Netherlands

⋆ marm3.14@gmail.com

## Abstract

As a new approach to efficiently describe correlation effects in the relativistic quantum world we propose to consider reduced density matrix functional theory, where the key quantity is the first-order reduced density matrix (1-RDM). In this work, we first introduce the theoretical foundations to extend the applicability of this theory to the relativistic domain. Then, using the so-called no-pair (np) approximation, we arrive at an approximate treatment of the relativistic effects by focusing on electronic wavefunctions and neglecting explicit contributions from positrons. Within the np approximation the theory becomes similar to the nonrelativistic case, with as unknown only the functional that describes the electron-electron interactions in terms of the 1-RDM. This requires the construction of functional approximations, and we therefore also present the relativistic versions of some common RDMFT approximations that are used in the nonrelativistic context and discuss their properties.

# 1   Introduction

When relativistic effects play a role in the quantum world, the Schrödinger equation must be replaced by the Dirac equation. This change of paradigm is required to describe the electronic structure of heavy elements in the periodic table [1–5] as electrons reach high velocities due to their strongly attractive nuclear potentials. When such elements are present, often (near) degeneracies in the electronic energies (e.g. due to spin-orbit coupling effects) occur and demand a good treatment of the so-called nondynamic (strong/static) correlation effects [6–14]. While methods like complete active space including second-order perturbation theory (CASPT2) [15–18], DMRG [19,20] and multi-reference coupled cluster theory [21–26] can be employed they all exhibit a high computational scaling with system size that limits their general applicability. An interesting efficient alternative is offered by reduced density matrix functional theory (RDMFT) as it has an intrinsic low-order scaling with system size [27,28].

RDMFT is emerging as a strong competitor to the widely used density functional theory (DFT) due to the possibility to use fractional occupation numbers, which facilitates the study of electronic systems where the so-called nondynamic correlation effects are enhanced [29–35]. Indeed, the popular working horse of physicists and chemists (i.e. the use of Kohn–Sham DFT approach with the usual density-functional approximations), in general, fails to account for nondynamic correlation effects [36] (except for a few cases like the approximation based on the fractional spin localized orbital scaling correction [37], some functional approximations developed using strictly correlated electrons [38], Becke's B13 functional [39,40], among others.). Actually, the capability of RDMFT to render nondynamic correlation effects has lead to a recent burst of this theory into new domains like the study of superconductivity [41] and of Bose–Einstein condensates [42]. In an attempt to extend it to the relativistic context, we validate its applicability by setting the theoretical foundations of relativistic RDMFT (ReRDMFT).

Already in 2002 Ohsaku et al. introduced a local (relativistic) quantum electro-dynamical (QED) RDMFT [43] theory in which the nonlocality properties of the first-order reduced density matrix (1-RDM) were not exploited and the so-called noninteracting kinetic energy needed

to be evaluated with an auxiliary noninteracting system. This complication may be avoided by using the full 1-RDM as all one-body interactions can then be evaluated as explicitly known functionals of the 1-RDM. In our work, we do therefore exploit also the nonlocality of the 1-RDM and introduce relativistic RDMFT (ReRDMFT). In this theory, we consider an external nonlocal potential (as it was employed by Gilbert in the nonrelativistic context [44]) and define the energy as a functional of the 1-RDM. In this way, the functional expression for all the one-body interactions is fully known in terms of this matrix and only the energy functional for electron-electron (as well as the positron-positron and the electron-positron) interactions remains unknown.

Recently, Toulouse proposed a relativistic density-functional theory based on a Fock-space effective QED Hamiltonian using the Coulomb or Coulomb–Breit two-particle interaction [45]. This theory, based on the works of Chaix et al. (see Refs. 46, 47), includes vacuum polarization effects through the creation of electron-positron pairs. Following his work we propose a similar approach called npvp-ReRDMFT that is capable to account for the effects of vacuum polarization within the no-pair vacuum-polarization approximation. Finally, neglecting the effect of vacuum polarization and assuming that a floating vacuum state is taken as reference each time spinor rotations are applied, we arrive to our last formulation called np-ReRDMFT. The np-ReRDMFT corresponds to the usual application of the no-pair approximation [48, 49].

This work is organized as follows: 1) we introduce the most general ReRDMFT approach, where creation and annihilation of electron-positron pairs is allowed. To that end, we first discuss the single-particle problem subject to a nonlocal external potential; then, we present its many-particle generalization and introduce the Fock space. Next, we analyze some properties of this Fock space and show how wavefunctions can be split according to the different charge sectors. Then, we focus on the effect of changing the basis representation that leads to vacuum polarization. Finally, we present the correlated wavefunction (that includes creation and annihilation of electron-positron pairs processes) and use it in combination with the constrained search formalism to introduce ReRDMFT. 2) As an approximation to the full ReRDMFT we include the so-called no-pair approximation at two levels. An initial approach where vacuum polarization effects are taken into account (leading to npvp-ReRDMFT) and a second approach where these effects are neglected (np-ReRDMFT). It is within the np-ReRDMFT framework that we evoke the Kramers' symmetry [50] and use it to adapt the nonrelativistic RDMFT functional approximations to the relativistic context. Finally, we discuss some of the properties of the functional approximations. Before proceeding, let us present Table 1 where we have collected the index conventions that will be employed in this work.

Table 1: Indices used throughout this work.

| Indices | labeling |
|---|---|
| $I, J, K, L$ | positive energy spinors (PS) |
| $R, S$ | negative energy spinors (NS) |
| $A, B$ | all spinors (NS $\cup$ PS) |
| $i, j, k, l$ | half of the PS (not related by Kramers' symmetry) |
| $\bar{i}, \bar{j}, \bar{k}, \bar{l}$ | half of the PS (Kramers partners of the unbarred spinors) |
| $\mu, \varepsilon, \tau, \eta$ | scalar orbitals (components of a 4-spinor) |
| $p, q$ | electron pairs |

## 2  ReRDMFT including electron-positron pair creation and annihilation processes.

### 2.1  The free particle Dirac equation and quantization of the Dirac field

Let us start by defining the time-independent free particle Dirac equation

$$\widehat{\mathbf{T}}_D(\mathbf{r})\psi_A(\mathbf{r}) = \left[-ic(\boldsymbol{\alpha}\cdot\boldsymbol{\nabla}_{\mathbf{r}}) + mc^2\boldsymbol{\beta}\right]\psi_A(\mathbf{r})$$
$$= E_A\psi_A(\mathbf{r}), \tag{1}$$

where $\widehat{\mathbf{T}}_D(\mathbf{r})$ is the usual first-quantized $4\times 4$ Dirac kinetic + rest mass operator, $i = \sqrt{-1}$, $c = 137.036$ a.u. is the speed of light, $m = 1$ a.u. is the electron mass,

$$\boldsymbol{\alpha} = (\boldsymbol{\alpha}_x, \boldsymbol{\alpha}_y, \boldsymbol{\alpha}_z)$$
$$= \left(\begin{pmatrix} \mathbf{0}_2 & \boldsymbol{\sigma}_x \\ \boldsymbol{\sigma}_x & \mathbf{0}_2 \end{pmatrix}, \begin{pmatrix} \mathbf{0}_2 & \boldsymbol{\sigma}_y \\ \boldsymbol{\sigma}_y & \mathbf{0}_2 \end{pmatrix}, \begin{pmatrix} \mathbf{0}_2 & \boldsymbol{\sigma}_z \\ \boldsymbol{\sigma}_z & \mathbf{0}_2 \end{pmatrix}\right), \tag{2}$$

$\mathbf{0}_2$ is the 2×2 null matrix,

$$\boldsymbol{\sigma}_x = \begin{pmatrix} 0 & 1 \\ 1 & 0 \end{pmatrix}, \qquad \boldsymbol{\sigma}_y = \begin{pmatrix} 0 & -i \\ i & 0 \end{pmatrix}, \qquad \boldsymbol{\sigma}_z = \begin{pmatrix} 1 & 0 \\ 0 & -1 \end{pmatrix}; \tag{3}$$

$$\boldsymbol{\beta} = \begin{pmatrix} \mathbb{I}_2 & \mathbf{0}_2 \\ \mathbf{0}_2 & -\mathbb{I}_2 \end{pmatrix}, \tag{4}$$

$\mathbb{I}_2$ is the 2×2 unit matrix. Solutions of the free Dirac equation ($\psi_A$) are 4-component-spinor orbitals

$$\psi_A(\mathbf{r}) = \begin{pmatrix} \phi_{A,1}(\mathbf{r}) \\ \phi_{A,2}(\mathbf{r}) \\ \phi_{A,3}(\mathbf{r}) \\ \phi_{A,4}(\mathbf{r}) \end{pmatrix}, \tag{5}$$

whose conjugate-transpose form reads

$$\psi_A^\dagger(\mathbf{r}) = \begin{pmatrix} \phi_{A,1}^*(\mathbf{r}) & \phi_{A,2}^*(\mathbf{r}) & \phi_{A,3}^*(\mathbf{r}) & \phi_{A,4}^*(\mathbf{r}) \end{pmatrix}. \tag{6}$$

Let us remark that the scalar functions ($\phi$) represent spatial orbitals because the spin is accounted by their arrangement in 4-component spinors. These solutions can be partitioned into a set of positive-energy 4-component spinors ($E_A > 0$) and a set of negative-energy ones ($E_A < 0$), i.e. $\{\psi_A\} = \{\psi_R\} \cup \{\psi_I\}$. From now on, we will designate the "positive energy spinors" ("negative energy spinors") as PS (NS). Hence, the Dirac field is quantized as

$$\widehat{\psi}(\mathbf{r}) = \sum_A \widehat{a}_A \psi_A(\mathbf{r}) = \sum_I \widehat{b}_I \psi_I(\mathbf{r}) + \sum_R \widehat{d}_R^\dagger \psi_R(\mathbf{r}), \tag{7}$$

where the sum has been decomposed into contributions involving electron(positron) annihilation(creation) operators. These operators obey the usual anticommutation relations

$$\{\widehat{a}_A, \widehat{a}_B^\dagger\} = \delta_{AB} \ \text{ and } \ \{\widehat{a}_A^\dagger, \widehat{a}_B^\dagger\} = \{\widehat{a}_A, \widehat{a}_B\} = 0. \tag{8}$$

## 2.2 The Hamiltonian operator including an external nonlocal potential, charge sectors in Fock space, and Bogoliubov transformations

Next, let us introduce a Hermitian nonlocal external potential that can be expressed in 4-component form as

$$
\mathbf{v}_{\text{ext}}^{\text{nl}}(\mathbf{r},\mathbf{r}') = \begin{pmatrix} v_{\text{ext},1,1}^{\text{nl}}(\mathbf{r},\mathbf{r}') & v_{\text{ext},1,2}^{\text{nl}}(\mathbf{r},\mathbf{r}') & v_{\text{ext},1,3}^{\text{nl}}(\mathbf{r},\mathbf{r}') & v_{\text{ext},1,4}^{\text{nl}}(\mathbf{r},\mathbf{r}') \\ v_{\text{ext},2,1}^{\text{nl}}(\mathbf{r},\mathbf{r}') & v_{\text{ext},2,2}^{\text{nl}}(\mathbf{r},\mathbf{r}') & v_{\text{ext},2,3}^{\text{nl}}(\mathbf{r},\mathbf{r}') & v_{\text{ext},2,4}^{\text{nl}}(\mathbf{r},\mathbf{r}') \\ v_{\text{ext},3,1}^{\text{nl}}(\mathbf{r},\mathbf{r}') & v_{\text{ext},3,2}^{\text{nl}}(\mathbf{r},\mathbf{r}') & v_{\text{ext},3,3}^{\text{nl}}(\mathbf{r},\mathbf{r}') & v_{\text{ext},3,4}^{\text{nl}}(\mathbf{r},\mathbf{r}') \\ v_{\text{ext},4,1}^{\text{nl}}(\mathbf{r},\mathbf{r}') & v_{\text{ext},4,2}^{\text{nl}}(\mathbf{r},\mathbf{r}') & v_{\text{ext},4,3}^{\text{nl}}(\mathbf{r},\mathbf{r}') & v_{\text{ext},4,4}^{\text{nl}}(\mathbf{r},\mathbf{r}') \end{pmatrix},
\tag{9}
$$

with the matrix elements obeying the constraint $v_{\text{ext},\mu,\varepsilon}^{\text{nl}}(\mathbf{r},\mathbf{r}') = \left[ v_{\text{ext},\varepsilon,\mu}^{\text{nl}}(\mathbf{r}',\mathbf{r}) \right]^{*}$ to ensure the hermiticity of the operator.

The bound particle Dirac equation that describes the states of a single particle (electron or positron) subject to a nonlocal external potential as in (9) reads

$$
\widehat{\mathbf{T}}_D(\mathbf{r})\psi_A(\mathbf{r}) + \int d\mathbf{r}' \mathbf{v}_{\text{ext}}^{\text{nl}}(\mathbf{r},\mathbf{r}')\psi_A(\mathbf{r}') = E_A \psi_A(\mathbf{r}).
\tag{10}
$$

The states $\psi$ depend on the choice of nonlocal external potential and form an orthonormal set ($\int d\mathbf{r}\psi_A^{\dagger}(\mathbf{r})\psi_B(\mathbf{r}) = \delta_{AB}$).

This single-particle equation can be readily generalized to noninteracting many-particle systems. To that end, let us recast the Hamiltonian to operate in Fock space as

$$
\begin{aligned}
\widehat{H}_0^{\nu} &= \widehat{T}_D + \widehat{V}_{\text{ext}}^{\text{nl}} \\
&= \int d\mathbf{r}d\mathbf{r}' \delta(\mathbf{r}-\mathbf{r}')\text{Tr}\left[ \widehat{\mathbf{T}}_D(\mathbf{r})\widehat{\mathbf{n}}_1(\mathbf{r},\mathbf{r}') \right] + \int d\mathbf{r}d\mathbf{r}' \text{Tr}\left[ \mathbf{v}_{\text{ext}}^{\text{nl}}(\mathbf{r}',\mathbf{r})\widehat{\mathbf{n}}_1(\mathbf{r},\mathbf{r}') \right] \\
&= \sum_{\mu,\tau} \int d\mathbf{r}d\mathbf{r}' \left[ \delta(\mathbf{r}-\mathbf{r}')T_{D,\mu,\tau}(\mathbf{r}) + v_{\text{ext},\mu,\tau}^{\text{nl}}(\mathbf{r}',\mathbf{r}) \right] \widehat{n}_{1,\tau,\mu}(\mathbf{r},\mathbf{r}'),
\end{aligned}
\tag{11}
$$

where we have introduced the one-particle density matrix operator whose elements read as

$$
\begin{aligned}
\widehat{n}_{1,\tau,\mu}(\mathbf{r},\mathbf{r}') &= \mathcal{N}\left[ \widehat{\psi}_{\mu}^{\dagger}(\mathbf{r}')\widehat{\psi}_{\tau}(\mathbf{r}) \right] \\
&= \sum_{I,J}\widehat{b}_I^{\dagger}\widehat{b}_J \phi_{I,\mu}^{*}(\mathbf{r}')\phi_{J,\tau}(\mathbf{r}) + \sum_{I}\sum_{S}\widehat{b}_I^{\dagger}\widehat{d}_S^{\dagger}\phi_{I,\mu}^{*}(\mathbf{r}')\phi_{S,\tau}(\mathbf{r}) \\
&\quad + \sum_{R}\sum_{J}\widehat{d}_R\widehat{b}_J \phi_{R,\mu}^{*}(\mathbf{r}')\phi_{J,\tau}(\mathbf{r}) - \sum_{R,S}\widehat{d}_S^{\dagger}\widehat{d}_R \phi_{R,\mu}^{*}(\mathbf{r}')\phi_{S,\tau}(\mathbf{r})
\end{aligned}
\tag{12}
$$

that is defined using creation and annihilation Dirac field operators with normal ordering [51, 52] (using Wick's theorem) $\mathcal{N}[\cdots]$ of the elementary operators [1] $\widehat{b}_I^{\dagger}$, $\widehat{d}_R^{\dagger}$, $\widehat{b}_I$, and $\widehat{d}_R$ w.r.t. the effective vacuum state ($|0_{\nu}\rangle$). The formal definition of the effective vacuum state requires a detailed discussion that we briefly summarize in the following lines. First we note that any Hamiltonian leads to a set of PS and NS. One could then define a bare vacuum state in which all spinors are unoccupied. This definition of the vacuum is problematic, however. Since NS are much lower in energy than PS, electrons occupying PS would be able to decay into empty NS by emitting photons resulting in an endless cascade of emission processes. To overcome this issue, Dirac proposed to fill all NS with electrons, introducing the so-called Dirac sea [53]. This is also problematic as the energy of such a vacuum state in which all NS are occupied is

---

[1] The spinor-components annihilation field operators are defined as $\widehat{\psi}_{\mu}(\mathbf{r}) = \sum_I \widehat{b}_I \phi_{I,\mu}(\mathbf{r}) + \sum_R \widehat{d}_R^{\dagger}\phi_{R,\mu}(\mathbf{r})$; the creation field operators are written as $\widehat{\psi}_{\mu}^{\dagger}(\mathbf{r}) = \sum_I \widehat{b}_I^{\dagger}\phi_{I,\mu}(\mathbf{r}) + \sum_R \widehat{d}_R \phi_{R,\mu}(\mathbf{r})$.

minus infinity. To avoid dealing with infinities, a reinterpretation was thus suggested in QED where all filled (by electrons) NS are redefined as empty positronic spinors. Thus, the effective vacuum state, which sets the zero of the energy scale, corresponds to the state where all PS do not contain electrons and all NS do not contain positrons. In the particular case when $\mathbf{v}_{\text{ext}}^{\text{nl}} = \mathbf{0}_{4\times4}$ ($\widehat{H}_0^\nu = \widehat{H}_0^0 \equiv \widehat{T}_D$), we refer to its effective vacuum state as $|0_0\rangle$ and to its spinor basis as $\boldsymbol{\psi}_A^0(\mathbf{r})$.

Let us assume that a given $\widehat{H}_0^\nu$ (with $\mathbf{v}_{\text{ext}}^{\text{nl}} \neq \mathbf{0}_{4\times4}$) is built initially in the free particle $\boldsymbol{\psi}_A^0(\mathbf{r})$ basis, with the normal ordering taken w.r.t. free particle vacuum $|0_0\rangle$. In this basis $\widehat{H}_0^\nu$ is not diagonal and it does not commute with the electron (positron) *number operators* $\widehat{N}_e = \sum_I \widehat{b}_I^\dagger \widehat{b}_I$ ($\widehat{N}_p = \sum_R \widehat{d}_R^\dagger \widehat{d}_R$). Moreover, $\widehat{H}_0^\nu$ does not commute with the total number of particles operator $\widehat{N} = \widehat{N}_e + \widehat{N}_p$. A useful operator that does commute with $\widehat{H}_0^\nu$ is, however, the *opposite charge operator* [49, 54] that is defined as

$$\widehat{Q} = \widehat{N}_e - \widehat{N}_p = \sum_I \widehat{b}_I^\dagger \widehat{b}_I - \sum_R \widehat{d}_R^\dagger \widehat{d}_R \tag{13}$$

(see Appendix A for more details). We then consider using the eigensolutions $\boldsymbol{\psi}_A^\nu$ that provide the diagonal representation $\widetilde{\widehat{H}_0^\nu}$ with the tilde symbol indicating that the Hamiltonian is transformed with respect to its original representation and has normal ordering w.r.t. its own vacuum $|0_\nu\rangle$. For consistency we will also indicate this vacuum as $|\widetilde{0}_\nu\rangle$ with the tilde symbol indicating that these quantities are transformed with respect to their original representations. In the current case $|\widetilde{0}_\nu\rangle = |0_\nu\rangle$, but this will change when two-particle interactions are considered below. In the $\widetilde{\boldsymbol{\psi}}_A(\mathbf{r}) = \boldsymbol{\psi}_A^\nu(\mathbf{r})$ basis the Hamiltonian commutes with the number of electrons (positrons) operators, the number of particles operator, and the opposite charge operator. Due to the diagonal representation, the pair creation and annihilation processes vanish; thence, the Hamiltonian conserves the particle number. Since both basis are orthonormal, they are related by a unitary transformation $\mathbf{V} = e^{\boldsymbol{\kappa}}$ [2], where the antihermitian matrix $\boldsymbol{\kappa}$ (formed by the parameters $\kappa_{AB} \in \mathbb{C}$) can be chosen to contain all its diagonal entries equal to zero to avoid redundant phase-shifts of the spinors. Therefore, the relationship between the two bases reads as

$$\widetilde{\boldsymbol{\psi}}_A(\mathbf{r}) = \sum_B \boldsymbol{\psi}_B^0(\mathbf{r}) V_{BA}. \tag{14}$$

The $\widetilde{\widehat{H}_0^\nu}$ operator is written in terms of the transformed operators ($\widetilde{\widehat{b}}_I^\dagger$, $\widetilde{\widehat{d}}_R^\dagger$, $\widetilde{\widehat{b}}_I$, and $\widetilde{\widehat{d}}_R$) that are given by

$$\widetilde{\widehat{a}}_A = e^{\widehat{\kappa}}\widehat{a}_A e^{-\widehat{\kappa}} = \sum_B \widehat{a}_B V_{BA}^*, \tag{15}$$

where the spinor rotation operator ($e^{\widehat{\kappa}}$) is expressed as the exponential of an antihermitian operator $\widehat{\kappa}$ with

$$\begin{aligned}\widehat{\kappa} &= \sum_{A,B} \kappa_{AB} \widehat{a}_A^\dagger \widehat{a}_B \\ &= \sum_{I,J} \kappa_{IJ} \widehat{b}_I^\dagger \widehat{b}_J + \sum_I \sum_S \kappa_{IS} \widehat{b}_I^\dagger \widehat{d}_S^\dagger + \sum_R \sum_J \kappa_{RJ} \widehat{d}_R \widehat{b}_J + \sum_{R,S} \kappa_{RS} \widehat{d}_R \widehat{d}_S^\dagger. \end{aligned} \tag{16}$$

Actually, the spinor rotation operator $e^{\widehat{\kappa}}$ corresponds to a Bogoliubov transformation mixing electron annihilation and positron creation operators [55, 56], producing modified electron

---

[2] We exploit the fact that any unitary matrix can be written in terms of an auxiliary antihermitian matrix $\boldsymbol{\kappa}$

and positron creation and annihilation operators that are consistent with a transformed vacuum.

Recalling that the Hamiltonian commutes with the opposite charge operator in any basis, it is convenient to classify its eigenstates within a Fock space that gathers together different particle-number sectors

$$\mathcal{F} = \bigoplus_{(N_e, N_p)=(0,0)}^{(\infty,\infty)} \mathcal{H}^{(N_e, N_p)} \ , \tag{17}$$

where $\bigoplus$ designates the direct sum. Alternatively, the Fock space can also be decomposed into charge ($Q = N_e - N_p$) sectors

$$\mathcal{F} = \bigoplus_{Q=-\infty}^{\infty} \mathcal{H}_Q. \tag{18}$$

For a given Hamiltonian $\widehat{H}_0^\nu$, the wavefunction of a system containing a fixed number of electron ($N_e$) and positrons ($N_p$) can be written as a single determinant (SD) [3]. The SD wavefunction reads as

$$|\Phi\rangle = \widehat{b}_{I_1}^\dagger \cdots \widehat{b}_{I_{N_e}}^\dagger \widehat{d}_{R_1}^\dagger \cdots \widehat{d}_{R_{N_p}}^\dagger |0_0\rangle, \tag{19}$$

and is is antisymmetric under the exchange of particles (as required for systems formed by fermions). Furthermore, the wavefunction in the basis where the Hamiltonian is diagonal can be written as a unitary rotation of the original wave function by operator $\widehat{V}$ as

$$
\begin{aligned}
|\widetilde{\Phi}\rangle = \widehat{V}|\Phi\rangle = e^{\widehat{\kappa}}|\Phi\rangle &= e^{\widehat{\kappa}} \widehat{b}_{I_1}^\dagger \cdots \widehat{b}_{I_{N_e}}^\dagger \widehat{d}_{R_1}^\dagger \cdots \widehat{d}_{R_{N_p}}^\dagger |0_0\rangle \\
&= e^{\widehat{\kappa}} \widehat{b}_{I_1}^\dagger e^{-\widehat{\kappa}} \cdots e^{\widehat{\kappa}} \widehat{b}_{I_{N_e}}^\dagger e^{-\widehat{\kappa}} e^{\widehat{\kappa}} \widehat{d}_{R_1}^\dagger e^{-\widehat{\kappa}} \cdots e^{\widehat{\kappa}} \widehat{d}_{R_{N_p}}^\dagger e^{-\widehat{\kappa}} e^{\widehat{\kappa}} |0_0\rangle \\
&= \widetilde{\widehat{b}}_{I_1}^\dagger \cdots \widetilde{\widehat{b}}_{I_{N_e}}^\dagger \widetilde{\widehat{d}}_{R_1}^\dagger \cdots \widetilde{\widehat{d}}_{R_{N_p}}^\dagger |\widetilde{0}_\nu\rangle,
\end{aligned} \tag{20}
$$

with $|\widetilde{0}_\nu\rangle = e^{\widehat{\kappa}}|0_0\rangle$. Let us also rewrite the annihilation Dirac field operators in the $\widetilde{\psi}_A(\mathbf{r})$ basis as

$$\widehat{\psi}(\mathbf{r}) = \sum_A \widetilde{\widehat{a}}_A \widetilde{\psi}_A(\mathbf{r}) = \sum_I \widetilde{\widehat{b}}_I \widetilde{\psi}_I(\mathbf{r}) + \sum_R \widetilde{\widehat{d}}_R^\dagger \widetilde{\psi}_R(\mathbf{r}), \tag{21}$$

a similar expression can be written for the creation Dirac field operator ($\widehat{\psi}^\dagger(\mathbf{r})$). In summary, as in the nonrelativistic context, the spinor rotation operator allows us to change the basis representation of the creation and annihilation operators, which may be used to obtain a diagonal representation of a particular Hamiltonian. Since the only difference between Hamiltonians is in the nonlocal external potential, this diagonal representation always depends on the given nonlocal external potential. Finally, taking the Taylor expansion of $e^{\widehat{\kappa}}$, it is easy to recognize that basis $\widetilde{\psi}_A(\mathbf{r})$ in which the Hamiltonian is diagonal is given by the set of $\{\kappa_{AB}\}$ parameters that lead to a stationary point [57] (i.e. a minimum, a maximum, or a saddle point) of the energy w.r.t. variations of these parameters.

## 2.3 Vacuum polarization

From now on, we will assume that an initially given orthonormal basis $\psi_A(\mathbf{r})$ is employed to express the Hamiltonian operator, this could be the $\psi_A^0(\mathbf{r})$ basis, but any orthonormal basis can in principle be used as long as it is possible to distinguish between initial positive and negative energy states, e.g. by diagonalizing an initial Hamiltonian. To stress this freedom of initial reference, we drop the symbol 0 from this definition as the choice of a free particle vacuum is

---

[3]Also known as the single Slater determinant wavefunction.

only one possibility. To avoid artefacts in the calculation of the kinetic energy due to basis set incompleteness, it is only necessary to build this initial basis with adequate kinetic balance [58–60] between the bases for the large and small components of the spinors. The initial effective vacuum state is now defined as $|0\rangle$ with the initial normal ordering taken w.r.t. this vacuum. The absence of tilde symbols on the basis functions and the vacuum hereby indicates that the Hamiltonian is non-diagonal in this basis. In the previous section we introduced the rotation operator ($e^{\widehat{\kappa}}$) and defined the transformed effective vacuum

$$|\widetilde{0}_\nu\rangle = e^{\widehat{\kappa}}|0\rangle \tag{22}$$

that provides a new reference for normal ordering the transformed operators. As we previously mentioned, the initially defined $\widehat{H}_0^\nu$ operator can be rewritten in another basis ($\widehat{\widetilde{H}_0^\nu}$) employing the operator $\widehat{\widetilde{n}}_{1,\tau,\mu}(\mathbf{r},\mathbf{r}')$ (i.e. expressing the one-body density matrix operator in terms of the elementary operators $\widetilde{\widehat{b}}^{\dagger}{}_I$, $\widetilde{\widehat{d}}^{\dagger}{}_R$, $\widetilde{\widehat{b}}_I$, and $\widetilde{\widehat{d}}_R$). Recalling that the normal ordering is introduced to guarantee that the energy of the effective vacuum is zero, i.e.,

$$\langle 0|\widehat{H}_0^\nu|0\rangle = \langle\widetilde{0}_\nu|\widehat{\widetilde{H}_0^\nu}|\widetilde{0}_\nu\rangle = 0, \tag{23}$$

when a Hamiltonian is not evaluated with its reference effective vacuum state it leads to nonzero contributions $\langle\widetilde{0}_\nu|\widehat{H}_0^\nu|\widetilde{0}_\nu\rangle \neq 0 \neq \langle 0|\widehat{\widetilde{H}_0^\nu}|0\rangle$. Actually, the relationship between the noninteracting Hamiltonian operators can simply be written as

$$\widehat{H}_0^\nu = \widehat{\widetilde{H}_0^\nu} + \widetilde{E}_0^0, \tag{24}$$

where

$$\widetilde{E}_0^0 = \langle\widetilde{0}_\nu|\widehat{H}_0^\nu|\widetilde{0}_\nu\rangle = \int d\mathbf{r}d\mathbf{r}'\mathrm{Tr}\left[\left(\delta(\mathbf{r}-\mathbf{r}')\widehat{\mathbf{T}}_D(\mathbf{r}) + \mathbf{v}_{\mathrm{ext}}^{\mathrm{nl}}(\mathbf{r}',\mathbf{r})\right)\widetilde{\mathbf{n}}_1^{\mathrm{vp}}(\mathbf{r},\mathbf{r}')\right], \tag{25}$$

is known as the vacuum polarization [46] (vp) energy (also present in the interacting particle picture, see below). And with the vacuum polarization one-particle density matrix defined as

$$\widetilde{\mathbf{n}}_1^{\mathrm{vp}}(\mathbf{r},\mathbf{r}') = \sum_R \widetilde{\psi}_R(\mathbf{r})\widetilde{\psi}_R^{\dagger}(\mathbf{r}') - \sum_R \psi_R(\mathbf{r})\psi_R^{\dagger}(\mathbf{r}') \tag{26}$$

whose components read as

$$\begin{aligned} \widetilde{n}_{1,\eta,\varepsilon}^{\mathrm{vp}}(\mathbf{r},\mathbf{r}') &= \langle\widetilde{0}_\nu|\widehat{n}_{1,\eta,\varepsilon}(\mathbf{r},\mathbf{r}')|\widetilde{0}_\nu\rangle \\ &= \sum_R \widetilde{\phi}_{R,\eta}(\mathbf{r})\widetilde{\phi}_{R,\varepsilon}^*(\mathbf{r}') - \sum_R \phi_{R,\eta}(\mathbf{r})\phi_{R,\varepsilon}^*(\mathbf{r}'). \end{aligned} \tag{27}$$

Before proceeding, let us remark that vp effects are present whenever spinor rotations mixing the positive and negative spinors are involved and the reference effective vacuum changes. When only one-body operators are present in the Hamiltonian, changes on it can be accounted for by orbital relaxation. This is no longer possible when also two-body operators are considered as will be discussed in the next section.

## 2.4 The interacting particle Hamiltonian and the configuration-interaction vacuum

In the more complete description of relativistic electronic dynamics provided by QED also the electromagnetic field is quantized such that the electron-electron (and electron-positron) interaction is mediated by the exchange of photons. This allows for a proper account of retardation effects due to the photons finite velocity. In the present work we take a simpler

approach and treat the electromagnetic field semi-classically with the approximate interaction provided by the Coulomb–Breit interaction. This operator corresponds to the single-photon exchange electron-electron scattering amplitude in QED evaluated with the zero-frequency limit of the photon propagator in the Coulomb electromagnetic gauge [54]. Unfortunately, this approximation makes the present theory not Lorentz invariant, but it is the most widely used interaction at present; it improves the description of the interaction w.r.t. nonrelativistic Coulomb interaction (i.e. $|\mathbf{r}_1 - \mathbf{r}_2|^{-1}$). Thus, the approximate interacting Hamiltonian reads as

$$
\begin{aligned}
\widehat{H} &= \widehat{H}_0^\nu + \widehat{W} \\
&= \widehat{H}_0^\nu + \frac{1}{2} \int d\mathbf{r}_1 d\mathbf{r}_2 \mathrm{Tr}\left[\mathbf{W}(\mathbf{r}_1,\mathbf{r}_2)\widehat{\mathbf{n}}_2(\mathbf{r}_1,\mathbf{r}_2)\right] \\
&= \widehat{H}_0^\nu + \frac{1}{2} \sum_{\mu,\varepsilon,\tau,\eta} \int d\mathbf{r}_1 d\mathbf{r}_2 W_{\mu,\varepsilon,\tau,\eta}(\mathbf{r}_1,\mathbf{r}_2)\widehat{n}_{2,\tau,\eta,\mu,\varepsilon}(\mathbf{r}_1,\mathbf{r}_2),
\end{aligned}
\tag{28}
$$

where

$$
W_{\mu,\varepsilon,\tau,\eta}(\mathbf{r}_1,\mathbf{r}_2) = \frac{1}{r_{12}}\left[\delta_{\mu,\tau}\delta_{\varepsilon,\eta} - \frac{1}{2}\left(\boldsymbol{\alpha}_{\mu,\tau}\cdot\boldsymbol{\alpha}_{\varepsilon,\eta}\right) - \frac{\left(\boldsymbol{\alpha}_{\mu,\tau}\cdot\mathbf{r}_{12}\right)\left(\boldsymbol{\alpha}_{\varepsilon,\eta}\cdot\mathbf{r}_{12}\right)}{2r_{12}^2}\right]
\tag{29}
$$

with $\mathbf{r}_{12} = \mathbf{r}_1 - \mathbf{r}_2$ and $r_{12} = |\mathbf{r}_{12}|$; and the pair-density matrix operator is defined using creation and annihilation Dirac field operators

$$
\widehat{n}_{2,\tau,\eta,\mu,\varepsilon}(\mathbf{r}_1,\mathbf{r}_2) = \mathcal{N}\left[\widehat{\psi}_\varepsilon^\dagger(\mathbf{r}_2)\widehat{\psi}_\mu^\dagger(\mathbf{r}_1)\widehat{\psi}_\tau(\mathbf{r}_1)\widehat{\psi}_\eta(\mathbf{r}_2)\right],
\tag{30}
$$

where the normal ordering is taken w.r.t. the effective vacuum state $|0\rangle$. Due to the interaction, a SD wavefunction (Eqs. (19) and (20)) is no-longer an eigenstate of $\widehat{H}$ and representations of $\widehat{H}$ are in general non-diagonal.

Let us start the search for the wavefunctions of $\widehat{H}$ by noticing that the Fock space can be split into charge sectors, such that the ground state energy of $\widehat{H}$ can be written as [4]

$$
E = \min_{|\Psi\rangle \in \mathcal{H}_Q} \langle\Psi|\widehat{H}_0^\nu + \widehat{W}|\Psi\rangle.
\tag{31}
$$

Thence, the wavefunctions must also be eigenstates of $\widehat{Q}$ with eigenvalue $Q$ (i.e. $\widehat{Q}|\Psi\rangle = Q|\Psi\rangle$). Consequently, $|\Psi\rangle$ can be constrained to have a given charge $Q$ (i.e. $\int d\mathbf{r}\langle\Psi|\mathrm{Tr}\left[\widehat{\mathbf{n}}_1(\mathbf{r},\mathbf{r})\right]|\Psi\rangle = Q$). Assuming that $\Psi$ is normalized to 1, a state that belongs to $\mathcal{H}_Q$ for a particular charge sector $Q \geq 0$ can be written (parameterized) as

$$
\begin{aligned}
|\Psi\rangle = \Bigg( &\sum_{I_1,\ldots,I_Q} c_{I_1\ldots I_Q}\widehat{b}_{I_1}^\dagger\cdots\widehat{b}_{I_Q}^\dagger + \sum_{I_1,\ldots,I_Q,I_{Q+1}}\sum_{R_1} c_{I_1\ldots I_Q I_{Q+1} R_1}\widehat{b}_{I_1}^\dagger\cdots\widehat{b}_{I_Q}^\dagger\widehat{b}_{I_{Q+1}}^\dagger\widehat{d}_{R_1}^\dagger \\
&+ \sum_{I_1,\ldots,I_Q,I_{Q+1},I_{Q+2}}\sum_{R_1,R_2} c_{I_1\ldots I_Q I_{Q+1} I_{Q+2} R_1 R_2}\widehat{b}_{I_1}^\dagger\cdots\widehat{b}_{I_Q}^\dagger\widehat{b}_{I_{Q+1}}^\dagger\widehat{b}_{I_{Q+2}}^\dagger\widehat{d}_{R_1}^\dagger\widehat{d}_{R_2}^\dagger + \cdots\Bigg)|0\rangle.
\end{aligned}
\tag{32}
$$

Since we account for all contributions to all orders there is no need for spinor rotations, i.e. spinor rotations become redundant, c.f. full configuration interaction (FCI) in the nonrelativistic context.

The charge sectors $Q < 0$ present the same structure as the $Q \geq 0$ ones. Thus, without any loss of generality and for practical reasons (i.e. as we are usually interested in electronic

---

[4] Assuming that the opposite charge operator ($\widehat{Q}$) and the interacting Hamiltonian operator ($\widehat{H}$) commute, which is only valid for an appropriate definition of the fermion-fermion interaction and with the adequate definition for the normal ordering.

wavefunctions more than in positronic ones), from now on will assume that our wavefunctions belong to the latter one.

The $Q = 0$ case is especially interesting as it could be used for a redefinition of the vacuum state. Up to now, we have considered an effective vacuum ($|0\rangle$) that could be obtained for some effective one-body potential $v$. This is not possible when we include an explicit two-electron interaction. Nevertheless, from the partition of the Fock space into charge sectors, it is possible to define instead a CI vacuum wavefunction ($|0^{\mathrm{CI}}\rangle$) as a linear combination of states with arbitrary number of electron-positron pairs that belongs to the charge sector $Q = 0$. Thus, the CI vacuum wavefunction reads as

$$|0^{\mathrm{CI}}\rangle = \left( c_0 + \sum_{I_1} \sum_{R_1} c_{I_1 R_1} \widehat{b}_{I_1}^\dagger \widehat{d}_{R_1}^\dagger + \sum_{I_1,I_2} \sum_{R_1,R_2} c_{I_1 I_2 R_1 R_2} \widehat{b}_{I_1}^\dagger \widehat{b}_{I_2}^\dagger \widehat{d}_{R_1}^\dagger \widehat{d}_{R_2}^\dagger + \cdots \right) |0\rangle, \qquad (33)$$

where spinor rotations are redundant because this wavefunction accounts for all vacuum contributions to all orders. Let us remark that although the two-electron operator does not change, these CI coefficients vary with the nonlocal external potential (e.g. when the molecular structure is changed). Only if this nonlocal external potential as well as the charge ($Q$) is kept fixed, then normal ordering [51, 52] can in principle be defined w.r.t. this $|0^{\mathrm{CI}}\rangle$ vacuum state. The vp effects (i.e. $\langle 0^{\mathrm{CI}}|\widehat{H}|0^{\mathrm{CI}}\rangle = 0$) can then be omitted. In the usual situation, when the external potential is allowed to change, vp effects cannot be neglected even when a CI vacuum is employed as the coefficients of CI vacuum depend on the external potential and this then affects the normal ordering.

In the nonrelativistic limit, when only electrons are considered in the wavefunction, the effective vacuum and the CI one coincide because in the reinterpretation of the NS the electrons are decoupled from the positrons and the latter do not contribute to the wavefunction. Moreover, in that limit the external potential does not affect the definition of the vacuum state; thence, the effective vacuum can be used independently of the interactions considered for the electrons and for any different external potential employed.

## 2.5 The 1-RDM and its natural orbital representation

Let us comment on some properties of the 1-RDM obtained from a wavefunction $|\Psi\rangle$ given by Eq. (32). Using the one-particle density matrix operator Eq. (12) we define the matrix elements of the 1-RDM in the spinor basis as

$$
\begin{aligned}
n_{1,\varepsilon,\mu}(\mathbf{r},\mathbf{r}') &= \langle \Psi | \widehat{n}_{1,\varepsilon,\mu}(\mathbf{r},\mathbf{r}') | \Psi \rangle \\
&= \sum_{I,J} {}^1D_I^J \phi_{I,\mu}^*(\mathbf{r}') \phi_{J,\varepsilon}(\mathbf{r}) + \sum_I \sum_R {}^1D_I^R \phi_{I,\mu}^*(\mathbf{r}') \phi_{R,\varepsilon}(\mathbf{r}) \\
&\quad + \sum_R \sum_I {}^1D_R^I \phi_{R,\mu}^*(\mathbf{r}') \phi_{I,\varepsilon}(\mathbf{r}) + \sum_{R,S} {}^1D_S^R \phi_{R,\mu}^*(\mathbf{r}') \phi_{S,\varepsilon}(\mathbf{r}),
\end{aligned}
\qquad (34)
$$

where we have introduced the 1-RDM coefficients ${}^1D_I^J = \langle \Psi | \widehat{b}_I^\dagger \widehat{b}_J | \Psi \rangle$, ${}^1D_I^R = \langle \Psi | \widehat{b}_I^\dagger \widehat{d}_R^\dagger | \Psi \rangle$, ${}^1D_R^I = \langle \Psi | \widehat{d}_R \widehat{b}_I | \Psi \rangle$, and ${}^1D_S^R = -\langle \Psi | \widehat{d}_S^\dagger \widehat{d}_R | \Psi \rangle$. In a nonrelativistic context, the coefficients arising from the expectation values $\langle \Psi | \widehat{b}_I^\dagger \widehat{d}_R^\dagger | \Psi \rangle$ and $\langle \Psi | \widehat{d}_R \widehat{b}_I | \Psi \rangle$ do not contribute since the wavefunction is formed by a fixed number of particles and due to the absence of positrons in this limit, the last contribution $\langle \Psi | \widehat{d}_S^\dagger \widehat{d}_R | \Psi \rangle$ then trivially vanishes.

Collecting the matrix elements we form the corresponding 4×4 matrix, and after trivial

algebra we obtain the full 1-RDM as

$$\mathbf{n}_1(\mathbf{r},\mathbf{r}') = \sum_{I,J} {}^1D_I^J \begin{pmatrix} \phi_{I,1}^*(\mathbf{r}')\phi_{J,1}(\mathbf{r}) & \phi_{I,2}^*(\mathbf{r}')\phi_{J,1}(\mathbf{r}) & \phi_{I,3}^*(\mathbf{r}')\phi_{J,1}(\mathbf{r}) & \phi_{I,4}^*(\mathbf{r}')\phi_{J,1}(\mathbf{r}) \\ \phi_{I,1}^*(\mathbf{r}')\phi_{J,2}(\mathbf{r}) & \phi_{I,2}^*(\mathbf{r}')\phi_{J,3}(\mathbf{r}) & \phi_{I,3}^*(\mathbf{r}')\phi_{J,2}(\mathbf{r}) & \phi_{I,4}^*(\mathbf{r}')\phi_{J,2}(\mathbf{r}) \\ \phi_{I,1}^*(\mathbf{r}')\phi_{J,3}(\mathbf{r}) & \phi_{I,2}^*(\mathbf{r}')\phi_{J,3}(\mathbf{r}) & \phi_{I,3}^*(\mathbf{r}')\phi_{J,3}(\mathbf{r}) & \phi_{I,4}^*(\mathbf{r}')\phi_{J,3}(\mathbf{r}) \\ \phi_{I,1}^*(\mathbf{r}')\phi_{J,4}(\mathbf{r}) & \phi_{I,2}^*(\mathbf{r}')\phi_{J,4}(\mathbf{r}) & \phi_{I,3}^*(\mathbf{r}')\phi_{J,4}(\mathbf{r}) & \phi_{I,4}^*(\mathbf{r}')\phi_{J,4}(\mathbf{r}) \end{pmatrix}$$

$$+ \sum_I \sum_R {}^1D_I^R (\cdots)_{4\times4} + \sum_R \sum_I {}^1D_R^I (\cdots)_{4\times4} + \sum_{R,S} {}^1D_S^R (\cdots)_{4\times4}$$

$$= \sum_{I,J} {}^1D_I^J \boldsymbol{\psi}_J(\mathbf{r})\boldsymbol{\psi}_I^\dagger(\mathbf{r}') + \sum_I \sum_R {}^1D_I^R \boldsymbol{\psi}_R(\mathbf{r})\boldsymbol{\psi}_I^\dagger(\mathbf{r}')$$

$$+ \sum_R \sum_I {}^1D_R^I \boldsymbol{\psi}_I(\mathbf{r})\boldsymbol{\psi}_R^\dagger(\mathbf{r}') + \sum_{R,S} {}^1D_S^R \boldsymbol{\psi}_S(\mathbf{r})\boldsymbol{\psi}_R^\dagger(\mathbf{r}'). \tag{35}$$

The trace of the 1-RDM is equal to the charge $Q$, i.e. $\sum_A {}^1D_A^A = Q$. Moreover, the coefficients ${}^1D_A^B$ form an Hermitian matrix that can be diagonalized, which leads to the natural orbital [5] (NO) representation of the 1-RDM

$$\mathbf{n}_1(\mathbf{r},\mathbf{r}') = \sum_A n_A \boldsymbol{\chi}_A(\mathbf{r})\boldsymbol{\chi}_A^\dagger(\mathbf{r}'), \tag{36}$$

where $\{n_A\}$ are known as the occupation numbers (ONs) that take values in the interval between $[-1,1]$ and the NOs are written as 4-component spinors $\boldsymbol{\chi}_A(\mathbf{r}) = \sum_B \boldsymbol{\psi}_B(\mathbf{r})U_{BA}$ (the columns of the $\mathbf{U}$ matrix contain the eigenvectors obtained from the diagonalization of the ${}^1\mathbf{D}$ matrix).

In the particular case of a SD wavefunction (19) the (canonical) $\boldsymbol{\psi}$ spinors coincide with the NOs with ONs that are then either 0 or $\pm1$. Consequently, the 1-RDM for $|\Phi\rangle$ reads as

$$\mathbf{n}_1^{SD}(\mathbf{r},\mathbf{r}') = \sum_I^{N_e} \boldsymbol{\psi}_I(\mathbf{r})\boldsymbol{\psi}_I^\dagger(\mathbf{r}') - \sum_R^{N_p} \boldsymbol{\psi}_R(\mathbf{r})\boldsymbol{\psi}_R^\dagger(\mathbf{r}'). \tag{37}$$

## 2.6 Introducing ReRDMFT through the constrained-search formalism

Considering the constrained-search formalism [61–63] we define the universal functional of the 1-RDM $W[\mathbf{n}_1]$ for $N$-representable 1-RDMs ($\mathbf{n}_1 \in \mathcal{D}_Q$) that come from a state $|\Psi\rangle \in \mathcal{H}_Q$ [6],

$$W[\mathbf{n}_1] = \min_{|\Psi\rangle \in \mathcal{H}_Q(\mathbf{n}_1)} \langle\Psi|\widehat{W}|\Psi\rangle$$

$$= \langle\Psi[\mathbf{n}_1]|\widehat{W}|\Psi[\mathbf{n}_1]\rangle, \tag{38}$$

where $\mathcal{H}_Q(\mathbf{n}_1)$ is the set of states $|\Psi\rangle \in \mathcal{H}_Q$ that yield a constrained 1-RDM ($\mathbf{n}_1$), and $|\Psi[\mathbf{n}_1]\rangle$ designates the state that minimizes this energy contribution [7]. Thence, we may write the ground state energy functional of the 1-RDM as

$$E_Q = \min_{\mathbf{n}_1 \in \mathcal{D}_Q} \left[ W[\mathbf{n}_1] + \int d\mathbf{r}d\mathbf{r}' \mathrm{Tr}\left[\left(\delta(\mathbf{r}-\mathbf{r}')\widehat{\mathbf{T}}_D(\mathbf{r}) + \mathbf{v}_{ext}^{nl}(\mathbf{r}',\mathbf{r})\right)\mathbf{n}_1(\mathbf{r},\mathbf{r}')\right] \right]. \tag{39}$$

---

[5]Could also be call as the natural spinor representation.

[6]We currently have no easy characterization of the set of $N$-representable 1-RDMs $\mathcal{D}_Q$. We at least know that the trace of these 1-RDMs should be equal to the charge $Q$. Other necessary conditions will be investigated in future work.

[7]Whether we have a true minimum, is an open question, but we suspect it to be the case like in the nonrelativistic limit [62].

At this point we would like to stress the similarity in structure of the expressions above with relativistic DFT, which is based on local potentials and the opposite charge density

$$n(\mathbf{r}) = \int d\mathbf{r}' \delta(\mathbf{r}-\mathbf{r}')\mathrm{Tr}\big[\mathbf{n}_1(\mathbf{r},\mathbf{r}')\big]. \tag{40}$$

The disadvantage of relativistic DFT is, however, that the free Dirac operator must also be included into the constrained search expression in this case, i.e.,

$$\begin{aligned}
F[n] &= \min_{|\Psi\rangle \in \mathcal{H}_Q(n)} \langle\Psi|\widehat{T}_D + \widehat{W}|\Psi\rangle \\
&= \langle\Psi[n]|\widehat{T}_D + \widehat{W}|\Psi[n]\rangle,
\end{aligned} \tag{41}$$

where $\mathcal{H}_Q(n)$ is the set of states $|\Psi\rangle \in \mathcal{H}_Q$ that yield a constrained opposite charge density $(n)$, and $|\Psi[n]\rangle$ designates the state that minimizes this energy contribution. Consequently, the energy functional within the constrained search formalism for $N$-representable opposite charge densities $(n \in \mathcal{D}_Q^{\mathrm{dens}})$ reads as

$$E_Q = \min_{n \in \mathcal{D}_Q^{\mathrm{dens}}} \left[ F[n] + \int d\mathbf{r} v_{\mathrm{ext}}(\mathbf{r}) n(\mathbf{r}) \right]. \tag{42}$$

Returning to ReRDMFT, clearly, the advantage of the functional of the 1-RDM over the one based on the opposite charge density is that all one-body interactions are an explicit functional of $\mathbf{n}_1$. For instance, adding an energy contribution arising from an external vector field ($\mathbf{A}^{\mathrm{ext}}$) to the above expression is straight forward: only the term $\delta(\mathbf{r}-\mathbf{r}')c\boldsymbol{\alpha}\cdot\mathbf{A}^{\mathrm{ext}}(\mathbf{r})$ needs to be added and fits the general form of $\mathbf{v}_{\mathrm{ext}}^{\mathrm{nl}}(\mathbf{r},\mathbf{r}')$ in (9). Thus, the functional of the 1-RDM contains more information than the functional of the current (i.e. $\mathbf{j}(\mathbf{r}) = c\boldsymbol{\psi}^\dagger(\mathbf{r})\boldsymbol{\alpha}\boldsymbol{\psi}(\mathbf{r}) = \mathrm{Tr}[c\boldsymbol{\alpha}\mathbf{n}_1(\mathbf{r},\mathbf{r})]$), which clearly shows the generality of the present theory and makes it interesting also for the description of magnetism in molecular systems. The functional presented in (39) establishes relativistic reduced density matrix functional theory (ReRDMFT). In particular, when the 1-RDM is given in the NO representation we will refer to it as relativistic natural orbital functional theory (ReNOFT). For practical purposes and to resemble nonrelativistic applications of RDMFT, we will use ReNOFT in this work when building functional approximations. An alternative approach to the constrained-search formalism is presented in the Appendix B, where we discuss the extension of Gilbert's theorem [44] to the relativistic domain.

## 3 The no-pair approximation and ReRDMFT

### 3.1 Vacuum polarization effects at fermion-fermion interacting case

The Hamiltonian given in Eq. (28) is written in normal-ordering w.r.t. to the initial effective vacuum state $|0\rangle$[8], which leads to $\langle 0|\widehat{H}|0\rangle = 0$. The relationship between the interacting Hamiltonian and its transformed counterpart (i.e. written in terms of transformed operators and basis) is given by the equation

$$\widehat{H} = \widehat{\widetilde{H}} + \widehat{\widetilde{V}}^{\mathrm{vp}} + \widetilde{E}_0, \tag{43}$$

---

[8]Recall that this vacuum state is given using some orthonormal basis $\psi_A(\mathbf{r})$ that is the initial guess. Also notice that the operators $\widehat{b}_I^\dagger$, $\widehat{d}_R^\dagger$, $\widehat{b}_I$, and $\widehat{d}_R$ are the initial ones.

where

$$
\begin{aligned}
\widehat{\widetilde{V}}^{\mathrm{vp}} &= \widehat{\widetilde{V}}_H^{\mathrm{vp}} + \widehat{\widetilde{V}}_{\mathrm{x}}^{\mathrm{vp}} \\
&= \sum_{\mu,\tau} \int d\mathbf{r}_1 \left[ \sum_{\varepsilon,\eta} \int d\mathbf{r}_2 W_{\mu,\varepsilon,\tau,\eta}(\mathbf{r}_1,\mathbf{r}_2) \widetilde{n}_{1,\eta,\varepsilon}^{\mathrm{vp}}(\mathbf{r}_2,\mathbf{r}_2) \right] \widehat{\widetilde{n}}_{1,\tau,\mu}(\mathbf{r}_1,\mathbf{r}_1) \\
&\quad - \sum_{\mu,\varepsilon,\tau,\eta} \int \int d\mathbf{r}_1 d\mathbf{r}_2 W_{\mu,\varepsilon,\tau,\eta}(\mathbf{r}_1,\mathbf{r}_2) \widetilde{n}_{1,\eta,\mu}^{\mathrm{vp}}(\mathbf{r}_2,\mathbf{r}_1) \widehat{\widetilde{n}}_{1,\tau,\varepsilon}(\mathbf{r}_1,\mathbf{r}_2),
\end{aligned}
\tag{44}
$$

$$
\widetilde{E}_0 = \langle \widetilde{0}_v | \widehat{H} | \widetilde{0}_v \rangle = \widetilde{E}_0^0 + \frac{1}{2} \int d\mathbf{r}_1 d\mathbf{r}_2 \operatorname{Tr}\left[ \mathbf{W}(\mathbf{r}_1,\mathbf{r}_2) \widetilde{\mathbf{n}}_2^{\mathrm{vp}}(\mathbf{r}_1,\mathbf{r}_2) \right],
\tag{45}
$$

where we have introduced the vp pair-density matrix (written in terms of its components) as

$$
\widetilde{n}_{2,\tau,\eta,\mu,\varepsilon}^{\mathrm{vp}}(\mathbf{r}_1,\mathbf{r}_2) = \widetilde{n}_{1,\eta,\varepsilon}^{\mathrm{vp}}(\mathbf{r}_2,\mathbf{r}_2) \widetilde{n}_{1,\tau,\mu}^{\mathrm{vp}}(\mathbf{r}_1,\mathbf{r}_1) - \widetilde{n}_{1,\tau,\varepsilon}^{\mathrm{vp}}(\mathbf{r}_1,\mathbf{r}_2) \widetilde{n}_{1,\eta,\mu}^{\mathrm{vp}}(\mathbf{r}_2,\mathbf{r}_1)
\tag{46}
$$

and $\tilde{E}_0^0$ was defined before in Eq. (25). Let us highlight that the definition of the vacuum as an effective vacuum ($|0\rangle$ or $|\widetilde{0}_v\rangle$, i.e. at the mean-field level), leads to only Hartree ($\widehat{\widetilde{V}}_H^{\mathrm{vp}}$) and exchange ($\widehat{\widetilde{V}}_x^{\mathrm{vp}}$) like terms for the interaction between the fermions (electrons or positrons) present in the $|\Psi\rangle$ and the polarization of the vacuum. Indeed, the $\widetilde{\mathbf{n}}_1^{\mathrm{vp}}$ can be regarded as the difference between the 1-RDM obtained from a SD wavefunction built from all positronic $\widetilde{\psi}$ states, i.e.,

$$
|\widetilde{\Phi}\rangle = \widehat{\widetilde{d}}_{R_1}^\dagger \cdots \widehat{\widetilde{d}}_{R_M}^\dagger |\widetilde{0}_v\rangle,
\tag{47}
$$

and the same kind of function build with all positronic $\psi$ states, i.e.,

$$
|\Phi\rangle = \widehat{d}_{R_1}^\dagger \cdots \widehat{d}_{R_M}^\dagger |0\rangle,
\tag{48}
$$

where $M$ is the total number of positronic states (i.e. the positronic states are fully occupied). Thus, the energy terms that account for the interaction between particles-vacuum and the 'vacuum-vacuum' effects, resemble the contribution arising from the frozen-core electrons in a multi-configuration self-consistent field calculations in the nonrelativistic context (e.g. complete active space self-consistent field). Consequently, only Hartree and exchange like terms are present.

The effective vacuum state $|\widetilde{0}_v\rangle$ that minimizes Eq. 45 can actually produce an energy that diverges to $-\infty$ due to infrared and ultraviolet divergences, which should be accounted for in a practical implementation of the present theory. Nevertheless, these issues go beyond the scope of the present work; therefore, we assume that a proper renormalization scheme is applied in order to keep everything finite in the rest of this work. Further details about infrared and ultraviolet divergences and how renormalization should be applied can be found in Refs. [45, 64–71].

Finally, let us recall that when the normal ordering is taken w.r.t. the CI vacuum state (for a fixed nonlocal external potential), vp effects do not need to be considered anymore even when spinor rotations are applied. In other words, only the CI vacuum leads to reference state that is invariant under spinor rotations. Such a definition of vacuum can, however, not be used in practice as the number of CI coefficients involved is formally infinite, which makes the CI expansion unmanageable from the practical perspective. Furthermore, working with finite basis sets still introduces a large number of CI coefficients (much larger than in non-relativistic Full CI calculations); thus, the use of the CI vacuum becomes rapidly intractable also for finite bases. In the following, we introduce the so-called no-pair approximation, where spinor rotations are required; therefore, they are always accompanied by vp energy contributions.

### 3.2 The no-pair approximation including vacuum polarization ReRDFMT approach

The no-pair (np) approximation [48,72] is a widely used simplification when dealing with relativistic calculations. It relies on the fact that most physical and certainly chemical processes correspond to much smaller energy exchanges (e.g. X-rays that interact with the electronic density contain ∼ 150keV) than required to create electron-positron pairs (> 1MeV) [73]. States that contain one or more positron pairs do therefore hardly contribute to the wave functions changes induced by such processes. Thus, in the description of common processes like chemical reactions or various kinds of spectroscopic phenomena, pure electronic wavefunctions are generally employed (i.e. $N_e = N$ and $N_p = 0$). Within the np approximation only the PS are populated while the NS remain empty. The np vp ground state energy reads as

$$E_N^{\text{npvp}} = \min_{|\Psi_+\rangle \in \widetilde{\mathcal{H}}^{(N,0)}} \langle \Psi_+ | \widehat{H} | \Psi_+ \rangle, \tag{49}$$

where the minimization is over normalized states in the set $\widetilde{\mathcal{H}}^{(N,0)} = e^{\widehat{\kappa}} \mathcal{H}^{(N,0)}$ that is the set of states generated by all spinor rotations of $N$-electron states. A state $|\Psi_+\rangle \in \widetilde{\mathcal{H}}^{(N,0)}$ can be written as

$$\begin{aligned}|\Psi_+\rangle &= e^{\widehat{\kappa}} \sum_{I_1,\dots,I_N} c_{I_1\dots I_N} \widehat{b}_{I_1}^\dagger \cdots \widehat{b}_{I_N}^\dagger |0\rangle \\ &= \sum_{I_1,\dots,I_N} c_{I_1\dots I_N} \widehat{\widetilde{b}}_{I_1}^\dagger \cdots \widehat{\widetilde{b}}_{I_N}^\dagger |\widetilde{0}_v\rangle.\end{aligned} \tag{50}$$

Since the state $|\Psi_+\rangle$ can also be written as $|\Psi_+\rangle = \widehat{\widetilde{P}}_+ |\Psi\rangle$, where $|\Psi\rangle \in \mathcal{H}_N$ is an arbitrary state constrained to have $N$ negative charges, and $\widehat{\widetilde{P}}_+$ projects onto the $N$-electron Hilbert space constructed by the $\{\widehat{\widetilde{b}}_I^\dagger\}$ operators. Thence, the energy minimization procedure given by Eq. (49) can also be regarded as an optimization w.r.t. the projector, where the energy minimization procedure implies finding an optimal vacuum state that depends on the $N_e$ through Eqs. (44) and (45). Lastly, using the constrained search formalism [61–63] and Eq. (49) we can readily introduce npvp-ReRDMFT, where vp effects are accounted at the effective QED level [9].

For making npvp-ReRDMFT more explicit, let us denote the np 1-RDM as

$$\mathbf{n}_1^+(\mathbf{r},\mathbf{r}') = \sum_{I,J} {}^1\widetilde{D}_I^J \widetilde{\psi}_J(\mathbf{r}) \widetilde{\psi}_I^\dagger(\mathbf{r}'), \tag{51}$$

where ${}^1\widetilde{D}_I^J = \langle \Psi_+ | \widehat{\widetilde{b}}_I^\dagger \widehat{\widetilde{b}}_J | \Psi_+ \rangle$. Making use of $\mathbf{n}_1^+$ and $\widetilde{\mathbf{n}}_1^{\text{vp}}$ all one-body interactions of Eq. (49) (i.e. the electronic and the vp effects) can be readily evaluated as

$$\langle \Psi_+ | \widehat{T}_D + \widehat{V}_{\text{ext}}^{\text{nl}} | \Psi_+ \rangle = \int d\mathbf{r} d\mathbf{r}' \text{Tr} \left[ \left( \delta(\mathbf{r}-\mathbf{r}') \widehat{T}_D(\mathbf{r}) + \mathbf{v}_{\text{ext}}^{\text{nl}}(\mathbf{r}',\mathbf{r}) \right) \left( \mathbf{n}_1^+(\mathbf{r},\mathbf{r}') + \widetilde{\mathbf{n}}_1^{\text{vp}}(\mathbf{r},\mathbf{r}') \right) \right]. \tag{52}$$

Also, the explicit vp contributions in terms of $\mathbf{n}_1^+$ and $\widetilde{\mathbf{n}}_1^{\text{vp}}$ to $\langle \Psi_+ | \widehat{W} | \Psi_+ \rangle$ is known through Eqs. (44) and (45). Indeed, only the pure electronic contribution to $\langle \Psi_+ | \widehat{\widetilde{W}} | \Psi_+ \rangle$ in terms of $\mathbf{n}_1^+$ is unknown and it needs to be approximated (see Appendix C for more details).

---

[9]Let us mention that NS are not populated within the np approximation, but they cannot be neglected when optimizing the PS (i.e. NS play a similar role as the secondary space in nonrelativistic complete active space calculations).

### 3.3 The no-pair approximation ReRDFMT approach

The vp contribution to the total energy (i.e. $\widetilde{E}_0$ and $\langle \Psi_+ | \widehat{\widetilde{V}}^{\mathrm{vp}} | \Psi_+ \rangle$) is usually neglected in practical applications of the np approximation. Redefining the normal ordering w.r.t. an effective vacuum state used whenever a spinor rotation is applied, the np energy reads

$$E_N^{\mathrm{np}} = \min_{|\Psi_+\rangle \in \widetilde{\mathcal{H}}^{(N,0)}} \langle \Psi_+ | \widehat{\widetilde{H}} | \Psi_+ \rangle, \tag{53}$$

where the Hamiltonian is written in normal ordering w.r.t. the floating vacuum state $|\widetilde{0}_v\rangle$. From Eq. (53) we may write, within the constrained-search formalism, the functional expression for $N$-representable np 1-RDMs ($\mathbf{n}_1^+ \in \mathcal{D}_N^+$) that come from a state $|\Psi_+\rangle \in \widetilde{\mathcal{H}}^{(N,0)}$

$$\begin{aligned} W^{\mathrm{np}}\left[\mathbf{n}_1^+\right] &= \min_{|\Psi_+\rangle \in \widetilde{\mathcal{H}}^{(N,0)}(\mathbf{n}_1^+)} \langle \Psi_+ | \widehat{\widetilde{W}} | \Psi_+ \rangle \\ &= \langle \Psi_+ \left[\mathbf{n}_1^+\right] | \widehat{\widetilde{W}} | \Psi_+ \left[\mathbf{n}_1^+\right] \rangle, \end{aligned} \tag{54}$$

where $\widetilde{\mathcal{H}}^{(N,0)}(\mathbf{n}_1^+)$ is the set of states $|\Psi_+\rangle \in \widetilde{H}^{(N,0)}$ that yield a constrained 1-RDM ($\mathbf{n}_1^+$), and $|\Psi_+\left[\mathbf{n}_1^+\right]\rangle$ designates the state that minimizes this energy contribution. Let us remark that we have explicitly written the Hamiltonian operator as a functional of the np 1-RDM to highlight that it changes during the minimization procedure. Actually, during the optimization procedure the gap between the floating vacuum state and the $N$-electron ground state energy is increased.

Consequently, the total energy functional that designates np-ReRDMFT can be defined as

$$E_N^{\mathrm{np}} = \min_{\mathbf{n}_1^+ \in \mathcal{D}_N^+} \left[ W^{\mathrm{np}}\left[\mathbf{n}_1^+\right] + \int d\mathbf{r} d\mathbf{r}' \mathrm{Tr}\left[ \left( \delta(\mathbf{r}-\mathbf{r}')\widehat{\mathbf{T}}_D(\mathbf{r}) + \mathbf{v}_{\mathrm{ext}}^{\mathrm{nl}}(\mathbf{r}',\mathbf{r}) \right) \mathbf{n}_1^+(\mathbf{r},\mathbf{r}') \right] \right]. \tag{55}$$

The np approximation resembles the nonrelativistic result because the trace of the $\mathbf{n}_1^+$ matrix is equal to the number of electrons, i.e. $\mathrm{Tr}\left[ {}^1\widetilde{\mathbf{D}} \right] = N_e$. The $\mathbf{n}_1^+$ matrix can also be expressed in the NO representation,

$$\mathbf{n}_1^+(\mathbf{r},\mathbf{r}') = \sum_I n_I \widetilde{\boldsymbol{\chi}}_I(\mathbf{r}) \widetilde{\boldsymbol{\chi}}_I^{\dagger}(\mathbf{r}'), \tag{56}$$

with $\sum_I n_I = N_e$ which allows us to introduce np-ReNOFT.

To conclude this section, as it is mentioned in Ref. [45], the np approximation regains the concept of an $N$-electron wavefunction,

$$\Psi_+(\mathbf{r}_1, \mathbf{r}_2, \ldots, \mathbf{r}_N) = \sum_{I_1 < \ldots < I_N} c_{I_1 \ldots I_N} \widetilde{\psi}_{I_1}(\mathbf{r}_1) \wedge \cdots \wedge \widetilde{\psi}_{I_N}(\mathbf{r}_N), \tag{57}$$

where $\wedge$ denotes the normalized antisymmetrized tensor product.

### 3.4 The $\langle \Psi_+ | \widehat{\widetilde{W}} | \Psi_+ \rangle$ term as an explicit functional of the second-order reduced density matrix.

To conclude this section, let us comment on some properties of the expectation value of the $\widehat{\widetilde{W}}$ operator. In the np approximation this term reads as

$$\langle \Psi_+ | \widehat{\widetilde{W}} | \Psi_+ \rangle = \sum_{I,J,K,L} {}^2D_{IJ}^{KL} \int d\mathbf{r}_1 d\mathbf{r}_2 \mathrm{Tr}\left[ \mathbf{W}(\mathbf{r}_1,\mathbf{r}_2)(\widetilde{\psi}_L(\mathbf{r}_1) \otimes \widetilde{\psi}_K(\mathbf{r}_2))(\widetilde{\psi}_I^{\dagger}(\mathbf{r}_2) \otimes \widetilde{\psi}_J^{\dagger}(\mathbf{r}_1)) \right] \tag{58}$$

where $\otimes$ denotes the tensor product, and ${}^2D_{IJ}^{KL} = \frac{1}{2}\langle\Psi_+|\widehat{\widetilde{b}}{}^\dagger_I\widehat{\widetilde{b}}{}^\dagger_J\widehat{\widetilde{b}}_L\widehat{\widetilde{b}}_K|\Psi_+\rangle$ is the np second-order reduced density matrix (2-RDM) element. Clearly, the $\langle\Psi_+|\widehat{\widetilde{W}}|\Psi_+\rangle$ contribution is an explicit functional of the 2-RDM. In the np approximation, the properties of the np 2-RDM are the same as in the nonrelativistic case. In particular, the trace of this matrix reads

$$\mathrm{Tr}\left[{}^2\mathbf{D}\right] = \sum_{I,J}{}^2D_{IJ}^{IJ} = \frac{N_e(N_e-1)}{2}. \tag{59}$$

Similarly, this matrix must fulfill the same $N$-representability conditions [74, 75] as in the nonrelativistic case, which we exploit in the following section to adapt/re-build some functional approximations for the relativistic problem. Finally, practical applications of relativistic quantum chemistry/physics normally retain only part of the contributions to $\widehat{\widetilde{W}}$. Indeed, the Coulomb–Breit interaction (29) consists of two physically distinct contributions: one representing the magnetic interaction between the electrons, and another one representing the retardation due to the finite velocity of the interaction [76]. The magnetic one was also discovered by Gaunt [77] and can be combined with the Coulomb interaction to yield

$$W_{\mu,\varepsilon,\tau,\eta}(\mathbf{r}_1,\mathbf{r}_2) = \frac{1}{r_{12}}\left[\delta_{\mu,\tau}\delta_{\varepsilon,\eta} - (\boldsymbol{\alpha}_{\mu,\tau}\cdot\boldsymbol{\alpha}_{\varepsilon,\eta})\right], \tag{60}$$

which will be used in the rest of this work (i.e. inserting it in Eq. (58)).

Before proceeding, let us stress that both energy expressions $E_N^{\mathrm{npvp}}$ and $E_N^{\mathrm{np}}$ can be minimized since the Hamiltonian is written using normal ordering. Hence, they are bounded from below and the state $|\Psi_+\rangle$ can actually be considered as a projected state from the $|\Psi\rangle\in\mathcal{H}_N$.

## 3.5 np-ReNOFT functional approximations by imposing Kramers' symmetry

Since the np approximation is the most common initial starting point for relativistic calculations, we propose the construction of functional approximations in this framework.

The explicit dependence of the $W\left[\mathbf{n}_1^+\right]$ functional in terms of the $\mathbf{n}_1^+$ is unknown (from now on we will drop the $^+$ super-index when referring to the np 1-RDM). In nonrelativistic NOFT the most accurate functionals [78–81] are built by imposing the so-called $N$-representability conditions [74,75,82] and using only up to two different indices to define 2-RDM elements, because the Hartree and the exchange contributions can be fully accounted for by using only two indices in the NO representation. These approximations usually employ a restricted approach, whose equivalent in the relativistic context corresponds to using Kramers' pairing symmetry when external magnetic fields are not included [76], which leads to a restricted formulation of spin magnetic moments. Then, a pair of NOs forms a Kramers' pair $(i,\bar{i})$ if they transform as $\widehat{\mathcal{K}}\widetilde{\boldsymbol{\chi}}_i = \widetilde{\boldsymbol{\chi}}_{\bar{i}}$ and $\widehat{\mathcal{K}}\widetilde{\boldsymbol{\chi}}_{\bar{i}} = -\widetilde{\boldsymbol{\chi}}_i$, where we have defined the time-reversal operator as

$$\widehat{\mathcal{K}} = -\mathrm{i}\begin{pmatrix}\boldsymbol{\sigma}_y & \mathbf{0}_2 \\ \mathbf{0}_2 & \boldsymbol{\sigma}_y\end{pmatrix}\widehat{\mathcal{K}}_0, \tag{61}$$

and where $\widehat{\mathcal{K}}_0$ is the complex conjugation operator. From now on we form two subsets of PS by splitting all spinors into Kramers' pairs [10]. Actually, the proper definition of Kramers' pairs is not unique because the pair $(i,\bar{i})$ is degenerate; thus, any unitary transformation applied to this pair leads to an equivalent pair of spinors that is still a Kramers' pair. In the non-relativistic limit, this corresponds exactly to the arbitrariness in the orientation of the spin quantization axis of the restricted formalism. These subsets are labeled with lowercase barred and unbarred indexes. Indeed, in practical calculations containing $M$ PS, we have $M/2$ Kramers' pairs and thus, $M/2$ (un)barred PS [11].

---

[10]Let us comment that the formation of pairs of degenerated spinor is not unique, but we have adopted the one given by Kramers' pairs in this work.

[11]The same kind of partitioning of the spinor states can be applied to the NS.

Imposing the Kramers' symmetry we may rewrite the $\widetilde{\widehat{H}}$ operator (including only the Coulomb and the Gaunt interactions) in the NOs representation as [54]

$$
\begin{aligned}
\widetilde{\widehat{H}}^{\mathrm{KR}} = {} & \sum_i h_{ii} \widehat{X}_{ii}^+ + \frac{1}{2} \sum_{i,j,k,l} \Big[ \langle ij|kl\rangle \widehat{x}_{ik,jl}^{++} - \langle ij|\boldsymbol{\alpha}\odot\boldsymbol{\alpha}|kl\rangle \widehat{x}_{ik,jl}^{--} + \langle \bar{i}j|kl\rangle \widehat{x}_{\bar{i}k,jl}^{++} - \langle \bar{i}j|\boldsymbol{\alpha}\odot\boldsymbol{\alpha}|kl\rangle \widehat{x}_{\bar{i}k,jl}^{--} \\
& + \langle ij|\bar{k}l\rangle \widehat{x}_{i\bar{k},jl}^{++} - \langle ij|\boldsymbol{\alpha}\odot\boldsymbol{\alpha}|\bar{k}l\rangle \widehat{x}_{i\bar{k},jl}^{--} \Big] + \frac{1}{4} \sum_{i,j,k,l} \Big[ \langle \bar{i}j|k\bar{l}\rangle \widehat{x}_{\bar{i}k,j\bar{l}}^{++} - \langle \bar{i}j|\boldsymbol{\alpha}\odot\boldsymbol{\alpha}|k\bar{l}\rangle \widehat{x}_{\bar{i}k,j\bar{l}}^{--} \Big] \quad (62) \\
& + \frac{1}{8} \sum_{i,j,k,l} \Big[ \langle \bar{i}\bar{j}|kl\rangle \widehat{x}_{ik,jl}^{++} - \langle \bar{i}\bar{j}|\boldsymbol{\alpha}\odot\boldsymbol{\alpha}|kl\rangle \widehat{x}_{ik,jl}^{--} + \langle ij|\bar{k}\bar{l}\rangle \widehat{x}_{i\bar{k},j\bar{l}}^{++} - \langle ij|\boldsymbol{\alpha}\odot\boldsymbol{\alpha}|\bar{k}\bar{l}\rangle \widehat{x}_{i\bar{k},j\bar{l}}^{--} \Big],
\end{aligned}
$$

where $\boldsymbol{\alpha}\odot\boldsymbol{\alpha} = \boldsymbol{\alpha}_x\otimes\boldsymbol{\alpha}_x + \boldsymbol{\alpha}_y\otimes\boldsymbol{\alpha}_y + \boldsymbol{\alpha}_z\otimes\boldsymbol{\alpha}_z$ and the integrals are defined as

$$
h_{II} = \int d\mathbf{r}d\mathbf{r}' \widetilde{\boldsymbol{\chi}}_I^\dagger(\mathbf{r}') \big( \delta(\mathbf{r}-\mathbf{r}')\widehat{\mathbf{T}}_D(\mathbf{r}) + \mathbf{v}_{\mathrm{ext}}^{\mathrm{nl}}(\mathbf{r},\mathbf{r}') \big) \widetilde{\boldsymbol{\chi}}_I(\mathbf{r}), \tag{63a}
$$

$$
\langle IJ|KL\rangle = \int d\mathbf{r}d\mathbf{r}' \frac{\big(\widetilde{\boldsymbol{\chi}}_I^\dagger(\mathbf{r})\otimes\widetilde{\boldsymbol{\chi}}_J^\dagger(\mathbf{r}')\big)\big(\widetilde{\boldsymbol{\chi}}_K(\mathbf{r})\otimes\widetilde{\boldsymbol{\chi}}_L(\mathbf{r}')\big)}{|\mathbf{r}'-\mathbf{r}|}, \tag{63b}
$$

$$
\langle IJ|\boldsymbol{\alpha}\odot\boldsymbol{\alpha}|KL\rangle = \int d\mathbf{r}d\mathbf{r}' \frac{\big(\widetilde{\boldsymbol{\chi}}_I^\dagger(\mathbf{r})\otimes\widetilde{\boldsymbol{\chi}}_J^\dagger(\mathbf{r}')\big)\big[\boldsymbol{\alpha}\odot\boldsymbol{\alpha}\big]\big(\widetilde{\boldsymbol{\chi}}_K(\mathbf{r})\otimes\widetilde{\boldsymbol{\chi}}_L(\mathbf{r}')\big)}{|\mathbf{r}'-\mathbf{r}|}, \tag{63c}
$$

$\widehat{X}_{ij}^s = (1+s\widehat{T}_{ij})\widehat{\bar{b}}_i^\dagger \widehat{\bar{b}}_j$, $\widehat{x}_{IK,JL}^{s_1 s_2} = (1+s_1\widehat{T}_{ik})(1+s_2\widehat{T}_{jl})\widehat{\bar{b}}_i^\dagger \widehat{\bar{b}}_J^\dagger \widehat{\bar{b}}_L \widehat{\bar{b}}_K = \widehat{x}_{JL,IK}^{s_2 s_1}$ with $\widehat{T}_{ij}\widehat{\bar{b}}_i^\dagger \widehat{\bar{b}}_j = \widehat{\bar{b}}_{\bar{j}}^\dagger \widehat{\bar{b}}_{\bar{i}}$ and $\widehat{T}_{ij}\widehat{\bar{b}}_{\bar{i}}^\dagger \widehat{\bar{b}}_j = -\widehat{\bar{b}}_{\bar{j}}^\dagger \widehat{\bar{b}}_i$ [12]. Let us remark that the $+$ and $-$ signs placed in Eq. (62) do not refer to positive or negative energy spinors; they only enter in the definition of $\widehat{X}_{IJ}^s$ and $\widehat{x}_{IJ,KL}^{s_1 s_2}$. Using the above Hamiltonian, the np energy reads as $E_N^{\mathrm{np}} = \langle \Psi_+|\widetilde{\widehat{H}}^{\mathrm{KR}}|\Psi_+\rangle$. For keeping the notation as concise as possible, we will drop the KR superscript on the Hamiltonian from now on.

Within nonrelativistic NOFT, most of the approximations rely on the usage of only up to two different indices in the electron-repulsion integrals to account for Hartree, exchange, and (some) correlation effects. Actually, retaining up to two indices most of the (nonrelativistic) NOFT approximations are able to retrieve the so-called nondynamic electron correlation energy, but they fail to account for the dynamic one (that can be accounted using different strategies [81, 83, 84]). Following the same strategy for np-ReNOFT and retaining only up to two different indices in Eqs. (63b) and (63c) the np energy in the NO representation, Eq. (55), reads as

$$
\begin{aligned}
E_N^{\mathrm{np}} \approx {} & \sum_i h_{ii}\big(n_i + n_{\bar{i}}\big) + \sum_{i,j}\Big({}^2D_{ij}^{ij} + {}^2D_{\bar{i}j}^{\bar{i}j} + {}^2D_{i\bar{j}}^{i\bar{j}} + {}^2D_{\bar{i}\bar{j}}^{\bar{i}\bar{j}}\Big)J_{ij} \\
& - \sum_{i,j}\Big({}^2D_{ij}^{ij} - {}^2D_{\bar{i}j}^{\bar{i}j} - {}^2D_{i\bar{j}}^{i\bar{j}} + {}^2D_{\bar{i}\bar{j}}^{\bar{i}\bar{j}}\Big)J_{ij}^G + \sum_{i,j}\Big[\Big({}^2D_{ij}^{ji} + {}^2D_{\bar{j}\bar{i}}^{\bar{i}\bar{j}}\Big)\big(K_{ij} - K_{ij}^G\big)\Big] \\
& + \frac{1}{2}\sum_{i,j}\Big[\Big({}^2D_{ij}^{j\bar{i}} + {}^2D_{\bar{i}j}^{i\bar{j}} + {}^2D_{j\bar{i}}^{\bar{i}j} + {}^2D_{i\bar{j}}^{\bar{j}i}\Big)\big(L_{ij} - L_{ij}^G\big)\Big] + \sum_{i\neq j}\Big[\Big({}^2D_{ii}^{j\bar{j}} + {}^2D_{jj}^{\bar{i}i}\Big)\big(K_{ij} + K_{ij}^G\big)\Big] \\
& - \frac{1}{2}\sum_{i\neq j}\Big[\Big({}^2D_{jj}^{i\bar{i}} + {}^2D_{ii}^{j\bar{j}} + {}^2D_{jj}^{\bar{i}i} + {}^2D_{ii}^{\bar{j}j}\Big)\big(L_{ij} + L_{ij}^G\big)\Big], \tag{64}
\end{aligned}
$$

where $J_{ij} = \langle ij|ij\rangle$, $J_{ij}^G = \langle ij|\boldsymbol{\alpha}\odot\boldsymbol{\alpha}|ij\rangle$, $K_{ij} = \langle ij|ji\rangle$, $K_{ij}^G = \langle ij|\boldsymbol{\alpha}\odot\boldsymbol{\alpha}|ji\rangle$, $L_{ij} = \langle \bar{i}j|j\bar{i}\rangle$ (notice that $L_{ii} = 0$ [76]), and $L_{ij}^G = \langle \bar{i}j|\boldsymbol{\alpha}\odot\boldsymbol{\alpha}|j\bar{i}\rangle$. Interestingly, some NOFT approximations (i.e.

---

[12] The uppercase indices that enter in the definition of $\widehat{x}_{IK,JL}^{s_1 s_2}$ are readily used in the creation and annihilation operators, but they are used as (lowercase) unbarred indices in $T_{ij}$ operator. Actually, the $T_{ij}$ is only defined with unbarred indices.

GNOF/PNOF$x$ approximations) employed in nonrelativistic calculations [79, 85] already use $L$ integrals to account for correlation effects among opposite-spin electrons.

The properties of the matrix elements of the np 2-RDM are the same as of the nonrelativistic 2-RDM ones (i.e. the matrix elements present the same symmetry and antisymmetry properties upon exchange of indices, they enter similarly in the evaluation of the $N$-representability conditions, etc.); hence, we have adopted the same strategy that is used in the nonrelativistic context to develop np(vp)-ReRDMFT functional approximations. Indeed, keeping only up to two different indices and using the natural orbital basis permits us to account for all Hartree and exchange interactions (now also including the exchange among 'opposite spin' terms in np(vp)-ReNOFT[np(vp)-ReRDMFT]). But, it also facilitates the search for analytic constraints on the approximated 2-RDM matrix elements upon evaluation of the D-, Q-, and G-$N$-representability conditions. Indeed, using the same strategy as in the nonrelativistic context leads to relativistic functional approximations that correspond to a generalization of their nonrelativistic counterparts (like the Dirac–Hartree–Fock method represents a generalization of the nonrelativistic Hartree–Fock method). Consequently, with this strategy we can also ensure that the correct nonrelativistic limit is retrieved with the functionals proposed in the following sections.

### 3.5.1 The Dirac–Hartree–Fock approximation

The np Dirac–Hartree–Fock (DHF) approximations uses a SD wavefunction to approximate $|\Psi_+\rangle$ (occupying only PS). Actually, the DHF energy can be readily written from Eq. (64) for 'spin-compensated' systems where $n_i = n_{\bar{i}}$ and defining the matrix elements of the 2-RDM with the SD approximation ($^2\mathbf{D}^{SD}$)

$$(^2D_{IJ}^{KL})^{SD} = \frac{n_I n_J}{2}(\delta_{IK}\delta_{JL} - \delta_{IL}\delta_{JK}). \tag{65}$$

This approximation produces the correct symmetry and antisymmetry properties of the 2-RDM [86]. Thence, the np DHF energy reads as

$$
\begin{aligned}
E^{np,DHF} &= 2\sum_i h_{ii} n_i + \sum_{i,j} n_i n_j \Big[ 2J_{ij} - \big(K_{ij} - K_{ij}^G\big) - \big(L_{ij} - L_{ij}^G\big)\Big] \\
&= 2\sum_i^{N_e/2} h_{ii} + \sum_{i,j}^{N_e/2} \Big[ 2J_{ij} - \big(K_{ij} - K_{ij}^G\big) - \big(L_{ij} - L_{ij}^G\big)\Big].
\end{aligned} \tag{66}
$$

In the last expression we have use $n_i = n_{\bar{i}} = 1$ for the PS that form the SD wavefunction, and 0 otherwise. An efficient algorithm to optimize this functional has been recently proposed by Sun et al. [87]. Moreover, this functional reduces to the one provided by Hafner for a purely Coulomb electron-electron interaction (i.e. $W_{\mu,\varepsilon,\tau,\eta}(\mathbf{r}_1, \mathbf{r}_2) = \frac{\delta_{\mu,\tau}\delta_{\varepsilon,\eta}}{r_{12}}$) from Ref. [88].

In the following, we introduce two paths to build ReNOFT approximations that are also used in the nonrelativistic context. The functionals presented in this work are proposed for 'spin-compensated' systems, i.e. the NOs forming Kramers' pairs always show the same ONs ($n_i = n_{\bar{i}}$). The first family of approximations is based on mimicking the exchange-correlation hole, while the second one on imposing $N$-representability conditions. Nevertheless, both paths approximate the 2-RDM elements as functions of the ONs, i.e.

$$^2D_{IJ}^{KL} = {}^2D_{IJ}^{KL}(n_I, n_J, n_K, n_L) \tag{67}$$

and does not take into account the actual shape of the spinors.

### 3.5.2  $f(n_I, n_J)$-functional approximations

The family of nonrelativistic $f(n_I, n_J)$-functional approximations is introduced as an attempt to separate the electron-electron interactions into Hartree and exchange-correlation effects [86]. These approximations account for exchange-correlation effects by approximating the so-called exchange-correlation hole. The exchange-correlation contribution is accounted replacing the product $n_I n_J$ in the second term on the r.h.s. of the DHF energy by a $f(n_I, n_J)$ function that attenuates the exchange; thus, accounts not only for exchange but also for exchange-correlation effects. In this context, the relativistic extension of the $f(n_I, n_J)$-functional approximations (i.e. the 2RDM elements) reads as

$$\left(^2D_{IJ}^{KL}\right)^{\text{approx.}} = \frac{n_I n_J}{2}\delta_{IK}\delta_{JL} - \frac{f(n_I, n_J)}{2}\delta_{IL}\delta_{JK}. \tag{68}$$

In Table 2 we have collected some $f(n_I, n_J)$ functions that lead to some representative functional approximations.

Table 2: Representative $f(n_I, n_J)$-functional approximations.

| Functional | $f(n_I, n_J)$ | case | Ref. |
|---|---|---|---|
| MBB | $\sqrt{n_I n_J}$ | all | [89–91] |
| Power | $(n_I n_J)^\alpha$ | all | [92–94] |
| GU | $\sqrt{n_I n_J}$ | $I \neq J$ | [95] |
| | $n_I n_J$ | otherwise | |
| MLSIC | $n_I n_J \frac{a_0 + a_1 n_I n_J}{1 + b_1 n_I n_J}$ | $I \neq J$ | [96] |
| | $n_I n_J$ | otherwise | |
| | $a_0 = 1298.78$, | | |
| | $a_1 = 35114.4$, and | | |
| | $b_1 = 36412.2$ | | |

The functionals given by Eq. (68) retrieve their nonrelativistic counterparts. Although, in nonrelativistic NOFT approximations the $f(n_I, n_J)$ functions only affect the 'same-spin' terms (i.e. $(^2D_{ij}^{ji})^{\text{approx.}}$ and $(^2D_{\bar{i}\bar{j}}^{\bar{j}\bar{i}})^{\text{approx.}}$); therefore, the 'opposite-spin' ones (i.e. $(^2D_{i\bar{j}}^{\bar{j}i})^{\text{approx.}}$ and $(^2D_{\bar{i}j}^{j\bar{i}})^{\text{approx.}}$) could be defined as $-\frac{n_i n_j}{2}$ (or using any other function), which may lead to an extended definition of $f(n_I, n_J)$-functional approximations in the relativistic scenario.

Let us briefly introduce the origin of the functionals presented in Table 2 that have their origin in the nonrelativistic context. The Müller, Buijse and Baerends (MBB) functional was introduced independently by Müller and by Buijse and Baerends [89–91], it produces an approximated functional which fulfills the sum rule, (i.e. $\text{Tr}\left[^2\mathbf{D}\right] = \frac{N_e(N_e-1)}{2}$). This functional was derived from the requirement of minimal violation of the Pauli principle and from the analysis of Fermi and Coulomb holes.

In the case of the Power functional, the $\alpha$ parameter was first proven to have to be $\alpha \geq 0.5565$ to produce admissible densities (i.e. solutions that are: stable with respect to the corresponding Euler equations, $N$-representable and whose electron-electron interaction energy satisfies the Lieb–Oxford bound [97]). This functional is mostly used in solid state physics and, according to Sharma et al. [93], the MBB functional overcorrelates the electrons and the Power functional mediates with the overcorrelation of the electrons by using a parameter $\alpha > 1/2$. In some applications, the bound $\alpha \geq 0.5565$ is not fixed and values like 0.53 are employed [98].

The Goedecker and Umrigar (GU) and the Marques and Lathiotakis (MLSIC) functionals are examples of approximations developed to remove the so-call self-interaction [95, 96] terms (i.e. they remove the nonphysical $^2D_{ii}^{ii}$ and $^2D_{\bar{i}\bar{i}}^{\bar{i}\bar{i}}$ elements). The former corrects these terms on the MBB functional, while the latter corrects a previous version of a similar functional also based on a Padé approximant expression. The MLSIC parameters were optimized to reproduce the G2 test.

The energy expression for the $f(n_I, n_J)$ functionals reads as

$$E^{\text{rel-}x} = 2\sum_i h_{ii} n_i + \sum_{i,j} 2n_i n_j J_{ij} - \sum_{i,j} f(n_i, n_j)\left[\left(K_{ij} - K_{ij}^G\right) + \left(L_{ij} - L_{ij}^G\right)\right], \quad (69)$$

with $x =$MBB, Power, GU, and MLSIC.

### 3.5.3 Approximations built imposing $N$-representability conditions

A second family of functionals approximations is based on the reconstruction of the 2RDM elements imposing the D, Q, G $N$-representability conditions [74, 79, 86, 99].

From Eq. (64), we recognize the 2-RDM matrix elements that must be approximated. Indeed, the 2-RDM matrix elements can be organized in blocks of the form

$$^2\mathbf{D} = \begin{pmatrix} \left(^2\mathbf{D}_{ij}^{ij}, {}^2\mathbf{D}_{ji}^{ij}\right) & 0 & 0 & 0 \\ 0 & \left(^2\mathbf{D}_{\bar{j}j}^{\bar{i}i}, {}^2\mathbf{D}_{\bar{i}j}^{\bar{i}j}\Big|_{i \neq j}\right) & \left(^2\mathbf{D}_{j\bar{j}}^{\bar{i}i}, {}^2\mathbf{D}_{ji}^{\bar{i}j}\Big|_{i \neq j}\right) & 0 \\ 0 & \left(^2\mathbf{D}_{\bar{j}j}^{i\bar{i}}, {}^2\mathbf{D}_{\bar{i}j}^{j\bar{i}}\Big|_{i \neq j}\right) & \left(^2\mathbf{D}_{j\bar{j}}^{i\bar{i}}, {}^2\mathbf{D}_{ji}^{j\bar{i}}\Big|_{i \neq j}\right) & 0 \\ 0 & 0 & 0 & \left(^2\mathbf{D}_{\bar{i}\bar{j}}^{\bar{i}\bar{j}}, {}^2\mathbf{D}_{\bar{j}\bar{i}}^{\bar{i}\bar{j}}\right) \end{pmatrix}. \quad (70)$$

We may identify three decoupled blocks containing different spinor combinations. The $\left(^2\mathbf{D}_{ij}^{ij}, {}^2\mathbf{D}_{ji}^{ij}\right)$ and $\left(^2\mathbf{D}_{\bar{i}\bar{j}}^{\bar{i}\bar{j}}, {}^2\mathbf{D}_{\bar{j}\bar{i}}^{\bar{i}\bar{j}}\right)$ blocks resemble the $\alpha\alpha$ and $\beta\beta$ blocks of the nonrelativistic case, while the large middle block corresponds to the $\alpha\beta\alpha\beta$, $\alpha\beta\beta\alpha$, etc. terms. In the nonrelativistic limit, the off-diagonal terms of the middle block do not contribute to the energy as they cancel upon integration over spin (e.g. the block $^2\mathbf{D}_{j\bar{j}}^{\bar{i}i} = 0$). Proposing functional approximations is facilitated by the introduction of the so-called cumulant matrix, which can be defined as

$$\Lambda = {}^2\mathbf{D} - {}^2\mathbf{D}^{\text{SD}}. \quad (71)$$

Then, the auxiliary matrices $\Delta$ and $\Pi$ are defined to approximate the cumulant matrix (as in the nonrelativistic context). These auxiliary matrices facilitate the evaluation of the $N$-representability conditions. Upon evaluation of the D-, Q-, and G- $N$-representability conditions (see the Appendix D for more details), and dividing the spinor space $\Omega$ into subspaces mutually disjoint $\Omega_p$ (see Fig. 1 for more details) we arrive to the relativistic version of the GNOF/PNOF$x$ ($x = 5, 7$) functional approximations [81, 83, 100] that can be written as

$$E^{\text{rel-GNOF/PNOF}x} = \sum_{p=1}^{N_e/2} E_p + \sum_{q \neq p}^{N_e/2} E_{qp}. \quad (72)$$

The first sum accounts for all intra-subspace contributions and reads as

$$E_p = \sum_{i \in \Omega_p} n_i(2h_{ii} + J_{ii} + J_{ii}^G + L_{ii}^G) + \sum_{\substack{i,j \in \Omega_p \\ i \neq j}} \Pi_{i,j}^{\text{intra}}(K_{ij} + K_{ij}^G + L_{ij} + L_{ij}^G), \quad (73)$$

where

$$\Pi_{i,j}^{\text{intra}} = \begin{cases} -\sqrt{n_i n_j}, & i \text{ or } j \le N_e/2 \\ +\sqrt{n_i n_j}, & i, j > N_e/2, \end{cases} \tag{74}$$

the second sum accounts for inter-subspace contributions ($E_{ba}$) that can be defined as

$$E_{qp} = \sum_{i\in\Omega_q}\sum_{j\in\Omega_p} n_i n_j \left[ 2J_{ij} - (K_{ij} - K_{ij}^G) - (L_{ij} - L_{ij}^G) \right] + \sum_{i\in\Omega_q}\sum_{j\in\Omega_p} \Pi_{i,j;p,q}^{\text{inter}}(K_{ij} + K_{ij}^G + L_{ij} + L_{ij}^G), \tag{75}$$

where $\Pi_{i,j;p,q}^{\text{inter}} = 0$ in rel-PNOF5, $\Pi_{i,j;p,q}^{\text{inter}} = -\sqrt{n_i h_i n_j h_j}$ in rel-PNOF7 [101], $\Pi_{i,j;p,q}^{\text{inter}} = -4n_i h_i n_j h_j$ in rel-PNOF7s [13][14], and

$$\Pi_{i,j;p,q}^{\text{inter}} \begin{cases} n_i^d n_j^d - \sqrt{n_i h_i n_j h_j} - \sqrt{n_i^d n_j^d}, & i > q, j = p \text{ or } i = q, j > p \\ n_i^d n_j^d - \sqrt{n_i h_i n_j h_j} + \sqrt{n_i^d n_j^d}, & i > q, j > p \\ 0, & i = q, j = p \end{cases} \tag{76}$$

with $n_i^d = \frac{n_i h_p^d}{h_p}$, $h_p = 1 - n_p$, and $h_p^d = h_p \exp\left[ -(h_p/(0.02\sqrt{2}))^2 \right]$ in GNOF [102]. Each electron pair is described by an even number of PS forming Kramers' pairs. Moreover, the rel-GNOF/PNOF$x$ functionals presented in this work produce 2-RDM elements that are real, but complex elements could be introduced by taking other functions to approximate the $\Pi^{\text{intra}}$ and $\Pi^{\text{inter}}$ sub-matrices.

As we did for the $f(n_I, n_J)$-functionals, we briefly introduce the origin of the nonrelativistic PNOF$x$ approximations whose relativistic adaptations are presented in this work for completeness. PNOF5 was developed to produce the correct electron distribution upon bond cleavage processes [100]. This functional was initially introduced in the perfect-pairing approach, where each sub-space $\Omega_p$ contained only two NOs. Although, it was later extended beyond the perfect-pairing approach to allow more orbitals in each sub-space. The lack of inter-subspace interaction in PNOF5 lead to the later versions of PNOF$x$ functionals (like PNOF7). The PNOF7 functional accounts for the interaction among pairs and has been proven account for the so-called nondynamic correlation energy, but it misses the dynamic one [34, 35, 101]. In an attempt to account for both types of electron correlation the GNOF functional was recently proposed [102]; it has proven to be capable of accounting for an important part of the so-called dynamic correlation energy. Though, it still misses important contributions that lead to weak interactions (like ones present in van der Waals interactions and Hydrogen bonds). Finally, let us mention that in the limit when the ONs tend to 0 and 1, all these functional approximations retrieve the HF energy expression. The same holds for their relativistic versions, that lead to the np DHF expression for the total energy (66) in this limit.

---

[13]A variant proposed for PNOF7 functional that is used to define NOF-MP2 method [81].

[14]The relativistic version of PNOF6 functional is not included because one of us proved that beyond the perfect-pairing approach this functional does not obey the sum rule [86]

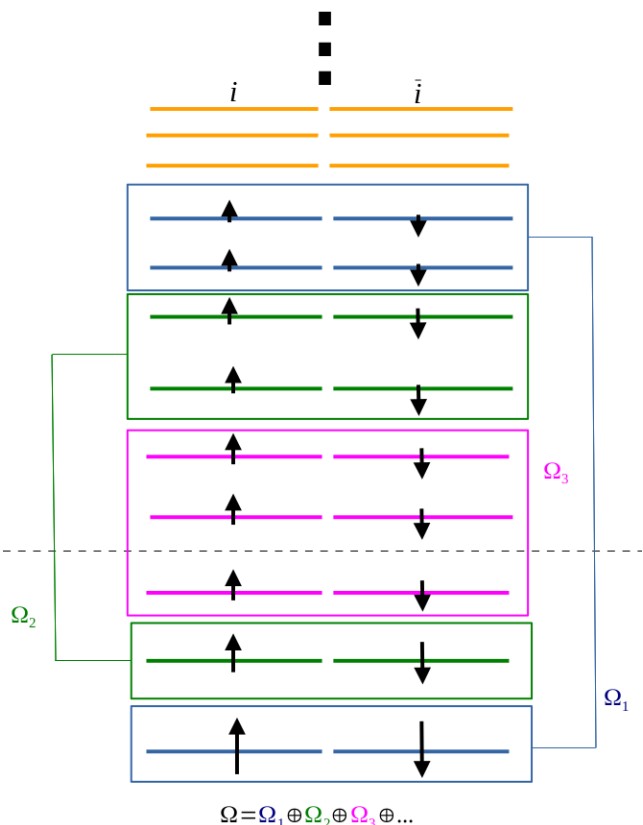

Figure 1: Partition of the spinor space $\Omega$ into subspaces mutually disjoint $\Omega_p$. While arrows refer to $\alpha$ or $\beta$ spin in the nonrelativistic context, in this work they correspond to electrons occupying spinors that are Kramers' pairs. Let us remark that some NOs may remain uncoupled (a.k.a. virtual spinors) like the orange ones in this scheme.

### 3.6 Properties of rel-GNOF/PNOF$x$ functionals.

For 'spin-compensated' two-electron systems (see Appendix E) the rel-GNOF/PNOF$x$ functionals introduced in this work reduce to

$$
\begin{aligned}
E_{2e-}^{\text{rel-GNOF/PNOF}x} = {} & \sum_i n_i(2h_{ii} + J_{ii} + J_{ii}^G + L_{ii}^G) - 2\sum_{\substack{i \\ i\neq 1}} \sqrt{n_1 n_i}(K_{1i} + K_{1i}^G + L_{1i} + L_{1i}^G) \\
& + \sum_{\substack{i,j \\ i\neq j, i\neq 1, j\neq 1}} \sqrt{n_i n_j}(K_{ij} + K_{ij}^G + L_{ij} + L_{ij}^G),
\end{aligned}
\tag{77}
$$

where the label $2e-$ indicates that it is for a two-electron system (also used in the Appendix E) and the PS are ordered in descending order w.r.t. their ON (and we have labeled as 1 the one with the largest ON). This functional can be proven to be the relativistic extension of the Fixed-Phases functional [103], which is based on the NO representation of the (singlet) wavefunction for two-electron systems [78] (see the Appendix E for more details).

The nonrelativistic PNOF5 energy functional expression is known to be equivalent to a constrained version of the energy expression of an antisymmetrized product of strongly-orthogonal geminals (APSG) wavefunction [51, 104]. In this section, we prove that the rel-PNOF5 functional is also equivalent to a (relativistic) APSG wavefunction within the np approximation.

To that end, let us introduce the $N$-electron wavefunction

$$\Psi_+^{\text{APSG}}(\mathbf{r}_1, \mathbf{r}_2, ..., \mathbf{r}_N) = \widehat{\mathcal{A}} \prod_{p=1}^{N_e/2} \Xi_p(\mathbf{r}_{2p-1}, \mathbf{r}_{2p}), \tag{78}$$

where

$$\Xi_p(\mathbf{r}, \mathbf{r}') = \frac{1}{\sqrt{2}} \sum_{i \in \Omega_p} c_i \left[ \widetilde{\chi}_i(\mathbf{r}) \otimes \widetilde{\chi}_{\bar{i}}(\mathbf{r}') - \widetilde{\chi}_i(\mathbf{r}') \otimes \widetilde{\chi}_{\bar{i}}(\mathbf{r}) \right] \tag{79}$$

is a geminal wavefunction written in the NO basis including phase factors (i.e. $c_i = \sqrt{n_i} e^{i\zeta_i}$, see Appendix E for more details), and the $\widehat{\mathcal{A}}$ operator produces the antisymmetrized product of geminal wavefunctions. The geminal wavefunctions are normalized $\int d\mathbf{r} d\mathbf{r}' \Xi_p^\dagger(\mathbf{r}, \mathbf{r}') \Xi_p(\mathbf{r}, \mathbf{r}') = 1$ for each electron pair, where

$$\Xi_p^\dagger(\mathbf{r}, \mathbf{r}') = \frac{1}{\sqrt{2}} \sum_{i \in \Omega_p} c_i^* \left[ \widetilde{\chi}_i^\dagger(\mathbf{r}) \otimes \widetilde{\chi}_{\bar{i}}^\dagger(\mathbf{r}') - \widetilde{\chi}_i^\dagger(\mathbf{r}') \otimes \widetilde{\chi}_{\bar{i}}^\dagger(\mathbf{r}) \right]. \tag{80}$$

Imposing the strong-orthogonality condition

$$\forall_{pq} \int d\mathbf{r}_1 d\mathbf{r}_2 \Xi_p^\dagger(\mathbf{r}_1, \mathbf{r}_2) \Xi_q(\mathbf{r}_1, \mathbf{r}_2) = \delta_{pq}, \tag{81}$$

we arrive to the following expression for the 1-RDM

$$\mathbf{n}_1^{+,\text{APSG}}(\mathbf{r}, \mathbf{r}') = \sum_{p=1}^{N_e/2} \sum_{i \in \Omega_p} n_i \left[ \widetilde{\chi}_i(\mathbf{r}) \widetilde{\chi}_i^\dagger(\mathbf{r}') + \widetilde{\chi}_{\bar{i}}(\mathbf{r}) \widetilde{\chi}_{\bar{i}}^\dagger(\mathbf{r}') \right]. \tag{82}$$

Thus, the relativistic APSG wavefunction permits us to write the energy functional

$$E^{\text{APSG}}[\{\zeta_i\}, \{n_i\}, \{\widetilde{\chi}_i\}] = 2 \sum_i n_i h_{ii} + \sum_{p \neq q}^{N_e/2} \sum_{\substack{i,j \\ i \in \Omega_p, j \in \Omega_q}} n_i n_j \left[ 2J_{ij} - (K_{ij} - K_{ij}^G) - (L_{ij} - L_{ij}^G) \right] \tag{83}$$

$$+ \sum_{p=1}^{N_e/2} \left[ \sum_{i \neq j \in \Omega_p} e^{i(\zeta_j - \zeta_i)} \sqrt{n_i n_j} (K_{ij} + K_{ij}^G + L_{ij} + L_{ij}^G) \right]$$

which can be optimized subject to the conditions: a) $\forall_i \ n_i \in [0, 1]$, and b) $N_e = 2\sum_i n_i$. Clearly, the energy functional given by Eq. (83) is a lower bound w.r.t. the rel-PNOF5 functional. Noticing that in rel-PNOF5 functional the phases $e^{i\zeta_i} = \pm 1$ are fixed (see Eq. (74)) while in $E^{\text{APSG}}$ they are parameters, we can readily conclude that in $E^{\text{APSG}}$ there is more flexibility during the optimization procedure that can lead to a lower energy.

## 4 Closing remarks

In this work we have introduced ReRDMFT at three different levels of theory. At the first level we have presented ReRDMFT including electron-positron pair creation/annihilation processes. At this level, the trace of the 1-RDM ($\mathbf{n}_1(\mathbf{r}, \mathbf{r}')$) is the charge $Q$; the minimum of the energy ($E_Q$) is attained for the CI ground state wavefunction ($|\Psi_Q\rangle$). This minimum is guaranteed due to the normal ordering procedure applied and it permits us to introduce ReRDMFT through the constrained search formalism. In principle, when the nonlocal external potential is fixed,

the normal ordering is taken w.r.t. the CI vacuum, and the total energy of the the system of interest is referenced w.r.t. this vacuum, its energy is invariant under spinor transformations (rotations). The energy as functional of the 1-RDM for all one-body interactions is presented, but functional approximations are required for the particle-particle interactions. The development of approximations for the particle-particle interactions will remain an open task, but a proper definition of these interactions would lead to a more precise description of the relativistic problem as it would also be valid for systems including positrons (e.g. electron-positron pairs).

Secondly, we have discussed the effect of taking the normal ordering w.r.t. the effective vacuum state $|\widetilde{0}_v\rangle$ and how this leads to the so-called vp when a spinor rotation is applied. These effects become unavoidable when the np approximation is adopted. Therefore, at the second level we have introduced npvp-ReRDMFT. We have discussed the importance of the NS spinors when spinor transformations are applied in the energy minimization procedure. At this level, two 1-RDMs play a crucial role (i.e. $\mathbf{n}_1^+(\mathbf{r}, \mathbf{r}')$ and $\widetilde{\mathbf{n}}_1^{\mathrm{vp}}(\mathbf{r}, \mathbf{r}')$). Within npvp-ReRDMFT, QED effects are taken at the mean-field level and the advantage of using a formulation based on the 1-RDMs becomes evident because all vp effects as well as electron-vacuum interactions are explicit functionals of $\mathbf{n}_1^+(\mathbf{r}, \mathbf{r}')$ and $\widetilde{\mathbf{n}}_1^{\mathrm{vp}}(\mathbf{r}, \mathbf{r}')$. Only the functional of the electron-electron interaction in terms of $\mathbf{n}_1^+(\mathbf{r}, \mathbf{r}')$ remains unknown. Fortunately, the functional approximations presented in this work (as part of the next level) are also valid for npvp-ReRDMFT.

Thirdly, the np-ReRDMFT is the last level of theoretical background introduced in this work. In np-ReRDMFT vp effects are neglected and a floating effective vacuum state is employed as reference. Within this framework the concept of an $N$-electron wavefunction is recovered. When the Hamiltonian preserves time-reversal symmetry it is possible to exploit Kramers' pairing symmetry; with Kramers' pairs we have written the energy expression using only up to two different indices. This expression allowed us to recognize the 2-RDM elements that need to be approximated as functions of the occupation numbers. In the end, the approximate 2-RDM elements proposed in this work were properly adapted from the two major families of functional approximations used in the nonrelativistic context. Subsequently, some properties of the functionals based on the inclusion of $N$-representability conditions were also discussed. The performance of np-ReRDMFT approximations is an open question that will be addressed in future works. Nevertheless, the computational cost of using np(vp)-ReRDFMT will increase w.r.t. its nonrelativistic counterpart, simply because we are forced to always use complex algebra and a spinor representation. The computational cost for the optimization should be similar to the cost required by a multiconfigurational-self-consistent field approach (specially for the optimization of the spinors). With the advantage that in np(vp)-ReRDMFT the optimization of the CI vector is replaced by the (usually cheaper) optimization over occupation numbers.

Finally, let us note that the analysis of the so-called generalized Pauli constraints [105–112] (GPC) of the ONs obtained when using some nonrelativistic functional approximations has attracted some attention [113] in the last few years. These conditions serve to approach to the possibility of obtaining pure $N$-representable 1-RDMs. They are particularly important in systems that are not 'spin-compensated' (i.e. that in our case do not preserve time-reversal symmetry [114–116]); thus, they are not contemplated in this work but they might be explored in future studies for treating 'spin-uncompensated' systems. Furthermore, pure $N$-representability conditions can also be imposed to the 2-RDM [116], and together with the D-, Q-, and G-conditions could also lead to improve results especially in strongly-correlated systems as Mazziotti suggested [116]; we may expect the same improvement for the relativistic functionals presented in this work.

## Acknowledgements

M.R.-M. acknowledges the European Commission for an Horizon 2020 Marie Skłodowska-Curie Individual Fellowship (891647-ReReDMFT). M.R.-M. would also like to acknowledge Prof. M. Piris for the fruitful discussions. K.J.H.G. acknowledges support by the Netherlands Organisation for Scientific Research (NWO) under Vici grant 724.017.001.

## A  Commutation relation between the Hamiltonian and the opposite charge operator

In order to prove the commutation relation between $\widehat{H}_0^\gamma$ and the opposite charge operator $\widehat{Q}$ let us first rewrite the Hamiltonian operator in the 4-component spinor basis as

$$\widehat{H}_0^\gamma = \sum_{I,J} \widehat{b}_I^\dagger \widehat{b}_J \mathcal{H}_{IJ} + \sum_I \sum_R \widehat{b}_I^\dagger \widehat{d}_R^\dagger \mathcal{H}_{IR} = \sum_{R,I} \widehat{d}_R \widehat{b}_I \mathcal{H}_{RI} - \sum_{R,S} \widehat{d}_S^\dagger \widehat{d}_R \mathcal{H}_{RS} \tag{84}$$

where

$$\mathcal{H}_{AB} = \int d\mathbf{r} d\mathbf{r}' \psi_A^\dagger(\mathbf{r}') \big[ \delta(\mathbf{r} - \mathbf{r}') \widehat{\mathbf{T}}_D + \mathbf{v}_{\text{ext}}^{\text{nl}}(\mathbf{r}', \mathbf{r}) \big] \psi_B(\mathbf{r}). \tag{85}$$

The evaluation of the commutator between opposite charge operator and first term of Hamiltonian lead us to

$$\sum_{I,J,K} \mathcal{H}_{IJ} \big[ \widehat{b}_I^\dagger \widehat{b}_J, \widehat{b}_K^\dagger \widehat{b}_K \big] - \sum_{I,J} \mathcal{H}_{IJ} \sum_R \big[ \widehat{b}_I^\dagger \widehat{b}_J, \widehat{d}_R^\dagger \widehat{d}_R \big] \tag{86}$$

$$= \sum_{I,J,K} \mathcal{H}_{IJ} \big( \widehat{b}_I^\dagger \widehat{b}_J \widehat{b}_K^\dagger \widehat{b}_K - \widehat{b}_K^\dagger \widehat{b}_K \widehat{b}_I^\dagger \widehat{b}_J \big)$$

$$= \sum_{I,J,K} \mathcal{H}_{IJ} \big( \widehat{b}_I^\dagger \widehat{b}_J \widehat{b}_K^\dagger \widehat{b}_K - \widehat{b}_I^\dagger \widehat{b}_J \widehat{b}_K^\dagger \widehat{b}_K + \widehat{b}_I^\dagger \widehat{b}_K \delta_{JK} - \widehat{b}_K^\dagger \widehat{b}_J \delta_{IK} \big)$$

$$= 0$$

the commutator $\big[ \widehat{b}_I^\dagger \widehat{b}_J, \widehat{d}_R^\dagger \widehat{d}_R \big] = 0$ because positronic and electronic creation and annihilation operators do commute. The proof that the fourth term of the Hamiltonian (i.e. $-\sum_{R,S} \widehat{d}_S^\dagger \widehat{d}_R \mathcal{H}_{RS}$) also commutes with the opposite charge operator follows closely the one presented for the first term; we therefore omit its detailed description. Next, the commutation relation between the second term of the Hamiltonian and the opposite charge operator reads as

$$\sum_{I,J} \sum_R \mathcal{H}_{IR} \big[ \widehat{b}_I^\dagger \widehat{d}_R^\dagger, \widehat{b}_J^\dagger \widehat{b}_J \big] - \sum_I \sum_{R,S} \mathcal{H}_{IR} \big[ \widehat{b}_I^\dagger \widehat{d}_R^\dagger, \widehat{d}_S^\dagger \widehat{d}_S \big] \tag{87}$$

$$\sum_{I,J} \sum_R \mathcal{H}_{IR} \big( \widehat{b}_I^\dagger \widehat{d}_R^\dagger \widehat{b}_J^\dagger \widehat{b}_J - \widehat{b}_J^\dagger \widehat{b}_J \widehat{b}_I^\dagger \widehat{d}_R^\dagger \big) - \sum_I \sum_{R,S} \mathcal{H}_{IR} \big( \widehat{b}_I^\dagger \widehat{d}_R^\dagger \widehat{d}_S^\dagger \widehat{d}_S - \widehat{d}_S^\dagger \widehat{d}_S \widehat{b}_I^\dagger \widehat{d}_R^\dagger \big)$$

$$\sum_{I,J} \sum_R \mathcal{H}_{IR} \big( \widehat{b}_I^\dagger \widehat{d}_R^\dagger \widehat{b}_J^\dagger \widehat{b}_J - \widehat{b}_I^\dagger \widehat{d}_R^\dagger \widehat{b}_J^\dagger \widehat{b}_J - \widehat{b}_J^\dagger \widehat{d}_R^\dagger \delta_{IJ} \big)$$

$$- \sum_I \sum_{R,S} \mathcal{H}_{IR} \big( \widehat{b}_I^\dagger \widehat{d}_R^\dagger \widehat{d}_S^\dagger \widehat{d}_S - \widehat{b}_I^\dagger \widehat{d}_R^\dagger \widehat{d}_S^\dagger \widehat{d}_S - \widehat{b}_I^\dagger \widehat{d}_S^\dagger \delta_{SR} \big)$$

$$= \sum_I \sum_R \mathcal{H}_{IR} \big( -\widehat{b}_I^\dagger \widehat{d}_R^\dagger + \widehat{b}_I^\dagger \widehat{d}_R^\dagger \big) = 0,$$

which proves that the second term also commutes with the Hamiltonian. Finally, the commutator between the third term of the Hamiltonian and the opposite charge operator leave us

with

$$\sum_{I,J}\sum_{R}\mathcal{H}_{RI}\left[\hat{d}_R\hat{b}_I,\hat{b}_J^\dagger\hat{b}_J\right]-\sum_{I}\sum_{R,S}\mathcal{H}_{RI}\left[\hat{d}_R\hat{b}_I,\hat{d}_S^\dagger\hat{d}_S\right] \tag{88}$$

$$\sum_{I,J}\sum_{R}\mathcal{H}_{RI}\left(\hat{d}_R\hat{b}_I\hat{b}_J^\dagger\hat{b}_J-\hat{b}_J^\dagger\hat{b}_J\hat{d}_R\hat{b}_I\right)-\sum_{I}\sum_{R,S}\mathcal{H}_{RI}\left(\hat{d}_R\hat{b}_I\hat{d}_S^\dagger\hat{d}_S-\hat{d}_S^\dagger\hat{d}_S\hat{d}_R\hat{b}_I\right)$$

$$\sum_{I,J}\sum_{R}\mathcal{H}_{RI}\left(\hat{d}_R\hat{b}_I\hat{b}_J^\dagger\hat{b}_J-\hat{d}_R\hat{b}_I\hat{b}_J^\dagger\hat{b}_J+\hat{d}_R\hat{b}_J\delta_{IJ}\right)-\sum_{I}\sum_{R,S}\mathcal{H}_{RI}\left(\hat{d}_R\hat{b}_I\hat{d}_S^\dagger\hat{d}_S-\hat{d}_R\hat{b}_I\hat{d}_S^\dagger\hat{d}_S+\hat{d}_S\hat{b}_I\delta_{RS}\right)$$

$$=\sum_{I}\sum_{R}\mathcal{H}_{RI}\left(\hat{d}_R\hat{b}_I-\hat{d}_R\hat{b}_I\right)=0,$$

that completes the demonstration of the commutation relation between the Hamiltonian and the opposite charge operator.

## B  Extending Gilbert's theorem to the relativistic domain.

In 1975, Gilbert extended the Hohenberg and Kohn theorems [117–119] for external nonlocal potentials. In this work we extend this theorem to the relativistic domain. Based on the existence of a lower bound of the energy (ensured by writing the creation an annihilation operators in normal ordering) we may define the ground state wavefunction $|\Psi\rangle \in \mathcal{H}_Q$ (see Eq. (32)) as the one that minimizes the energy for a particular Hamiltonian (Eq. (28)) and charge sector $Q$ of the Fock space.

We generalize Gilbert's theorem for relativistic nondegenerated ground states in the next theorem.

**Theorem 1** *Given a relativistic Hamiltonian $\widehat{H} = \widehat{T}_D + \widehat{V}_{ext}^{nl} + \widehat{W}$, whose nondegenerated ground state wavefunction reads as $\Psi_Q$ (for a particular charge sector $Q$). There exists a one-to-one correspondence between $\Psi_Q$ and its 1-RDM ($\mathbf{n}_1(\mathbf{r},\mathbf{r}')$).*

Proof. Assume that two-different nonlocal external potentials $\widehat{V}_{ext}^{1,nl}$ and $\widehat{V}_{ext}^{2,nl}$ that differ in more than a constant lead to the nondegenerated ground states $\Psi_Q^1$ and $\Psi_Q^2$ (for a charge sector $Q$). That is to say, $\widehat{H}_1\Psi_Q^1 = (\widehat{T}_D + \widehat{V}_{ext}^{1,nl} + \widehat{W})\Psi_Q^1 = E_1\Psi_Q^1$ and $\widehat{H}_2\Psi_Q^2 = E_2\Psi_Q^2$. The 1-RDMs of $\Psi_Q^1$ and $\Psi_Q^2$ are $\mathbf{n}_1^1(\mathbf{r},\mathbf{r}')$ and $\mathbf{n}_1^2(\mathbf{r},\mathbf{r}')$, respectively. If the two Hamiltonians only differ in the nonlocal external potential (i.e. $\widehat{H}_2 - \widehat{H}_1 = \widehat{V}_{ext}^{2,nl} - \widehat{V}_{ext}^{1,nl}$), by Rayleigh-Ritz variational principle we can write

$$E_1 = \langle\Psi_Q^1|\widehat{H}_1|\Psi_Q^1\rangle < \langle\Psi_Q^2|\widehat{H}_1|\Psi_Q^2\rangle = E_1^2 \tag{89}$$

and

$$E_2 = \langle\Psi_Q^2|\widehat{H}_2|\Psi_Q^2\rangle < \langle\Psi_Q^1|\widehat{H}_2|\Psi_Q^1\rangle = E_2^1, \tag{90}$$

so that

$$\Delta E = (E_1^2 - E_1) + (E_2^1 - E_2) > 0 \tag{91}$$

and

$$\begin{aligned}\Delta E &= (E_1^2 - E_1) + (E_2^1 - E_2)\\ &= \langle\Psi_Q^1|\widehat{H}_2 - \widehat{H}_1|\Psi_Q^1\rangle + \langle\Psi_Q^2|\widehat{H}_1 - \widehat{H}_2|\Psi_Q^2\rangle\\ &= -\int d\mathbf{r}d\mathbf{r}'\mathrm{Tr}\Big[\left(\mathbf{v}_{2,ext}^{nl}(\mathbf{r}',\mathbf{r}) - \mathbf{v}_{1,ext}^{nl}(\mathbf{r}',\mathbf{r})\right)\\ &\quad\times\left(\mathbf{n}_1^2(\mathbf{r},\mathbf{r}') - \mathbf{n}_1^1(\mathbf{r},\mathbf{r}')\right)\Big] > 0\,.\end{aligned} \tag{92}$$

Since we imposed that $\widehat{V}_{\text{ext}}^{1,\text{nl}}$ and $\widehat{V}_{\text{ext}}^{2,\text{nl}}$ are different, the corresponding 1-RDMs ($\mathbf{n}_1^1(\mathbf{r},\mathbf{r}')$ and $\mathbf{n}_1^2(\mathbf{r},\mathbf{r}')$) must also be different. Therefore, there exists a one-to-one correspondence between the 1-RDM and the wavefunction.

$$\Psi_Q \longleftrightarrow \mathbf{n}_1(\mathbf{r},\mathbf{r}'). \qquad \square \tag{93}$$

Consequently, there exists a functional of the energy

$$E_Q = E\left[\mathbf{n}_1\right] \qquad \text{for} \qquad \mathbf{n}_1(\mathbf{r},\mathbf{r}') \in \mathcal{D}_Q. \tag{94}$$

This functional retrieves its nonrelativistic counterpart for a fixed number of electrons (i.e. $Q = N_e$, see below). Furthermore, this functional leads to the existence of ReRDMFT and it complements the functional introduced through the constrained-search formalism.

**The no-pair vacuum polarization framework**

It is possible to establish a one-to-one correspondence between $|\Psi_+\rangle$ and $\mathbf{n}_1^+(\mathbf{r},\mathbf{r}')$ at the no-pair vacuum polarization approximation level of theory. Using the same interacting Hamiltonian operator but restricting the wavefunctions to be given by Eq. (50) (or Eq. (57)), similar arguments to the above presented lead to the desired one-to-one correspondence (i.e. $|\Psi_+\rangle \longleftrightarrow \mathbf{n}_1^+(\mathbf{r},\mathbf{r}')$). Finally, neglecting vacuum polarization effects, we may obtain the energy functional

$$E_{N_e} = E\left[\mathbf{n}_1^+\right] \qquad \text{for} \qquad \mathbf{n}_1 \in \mathcal{D}_{N_e}^+ \tag{95}$$

that allows us to introduce np-ReRDMFT and clearly retrieves the nonrelativistic limit.

## C The npvp energy functional

The $E^{\text{npvp}}$ functional of the np 1-RDM and the vp 1-RDM reads as

$$E^{\text{npvp}}\left[\mathbf{n}_1^+, \widetilde{\mathbf{n}}_1^{\text{vp}}\right] = \int d\mathbf{r}d\mathbf{r}'\text{Tr}\left[\left(\delta(\mathbf{r}-\mathbf{r}')\widehat{\mathbf{T}}_D(\mathbf{r}) + \mathbf{v}_{\text{ext}}^{\text{nl}}(\mathbf{r}',\mathbf{r})\right)\left(\mathbf{n}_1^+(\mathbf{r},\mathbf{r}') + \widetilde{\mathbf{n}}_1^{\text{vp}}(\mathbf{r},\mathbf{r}')\right)\right] \tag{96}$$

$$+ \sum_{\mu,\tau}\int d\mathbf{r}_1\left[\sum_{\varepsilon,\eta}\int d\mathbf{r}_2 W_{\mu,\varepsilon,\tau,\eta}(\mathbf{r}_1,\mathbf{r}_2)\widetilde{n}_{1,\eta,\varepsilon}^{\text{vp}}(\mathbf{r}_2,\mathbf{r}_2)\right]\widetilde{n}_{1,\tau,\mu}^+(\mathbf{r}_1,\mathbf{r}_1)$$

$$- \sum_{\mu,\varepsilon,\tau,\eta}\int\int d\mathbf{r}_1 d\mathbf{r}_2 W_{\mu,\varepsilon,\tau,\eta}(\mathbf{r}_1,\mathbf{r}_2)\widetilde{n}_{1,\eta,\mu}^{\text{vp}}(\mathbf{r}_2,\mathbf{r}_1)\widetilde{n}_{1,\tau,\varepsilon}^+(\mathbf{r}_1,\mathbf{r}_2)$$

$$+ \frac{1}{2}\int d\mathbf{r}_1 d\mathbf{r}_2 \text{Tr}\left[\mathbf{W}(\mathbf{r}_1,\mathbf{r}_2)\widetilde{\mathbf{n}}_2^{\text{vp}}(\mathbf{r}_1,\mathbf{r}_2)\right] + \widetilde{W}\left[\mathbf{n}_1^+\right],$$

where the explicit form of $\widetilde{W}\left[\mathbf{n}_1^+\right] = \langle\Psi_+\left[\mathbf{n}_1^+\right]|\widehat{\widetilde{W}}|\Psi_+\left[\mathbf{n}_1^+\right]\rangle$ is unknown and needs to be approximated. Finally, the usual np approximation corresponds to $E^{\text{np}}\left[\mathbf{n}_1^+\right] = E^{\text{npvp}}\left[\mathbf{n}_1^+,\mathbf{0}\right]$.

## D The $N$-representability conditions within the np approximation.

The so-called $N$-representability conditions of the 2-RDM aim to ensure that this matrix is associated to a wavefunction (i.e. $^2\mathbf{D} \longleftrightarrow |\Psi_+\rangle$). They also serve to propose a systematic way to build approximations for the 2-RDM matrix elements in terms of the 1-RDM ones. Following

the reconstruction procedure proposed in Ref. [79], let us define the matrix elements of the cumulant matrix as [15]

$$\lambda_{ij,kl} = -\frac{\Delta_{ij}}{2}\delta_{ik}\delta_{jl} + \frac{\Delta_{ij}}{2}\delta_{il}\delta_{jk}, \tag{97a}$$

and for the middle block

$$\lambda_{\bar{i}j,\bar{k}l} = -\frac{\Delta_{\bar{i}j}}{2}\delta_{ik}\delta_{jl} + \frac{\Pi_{\bar{i}i,\bar{k}k}}{2}\delta_{ij}\delta_{kl}, \tag{97b}$$

$$\lambda_{i\bar{j},k\bar{l}} = -\frac{\Delta_{i\bar{j}}}{2}\delta_{ik}\delta_{jl} + \frac{\Pi_{i\bar{i},k\bar{k}}}{2}\delta_{ij}\delta_{kl}, \tag{97c}$$

$$\lambda_{i\bar{j},\bar{k}l} = \frac{\Delta_{i\bar{j}}}{2}\delta_{il}\delta_{jk} - \frac{\Pi_{i\bar{i},\bar{k}k}}{2}\delta_{ij}\delta_{kl}, \tag{97d}$$

and

$$\lambda_{\bar{i}j,k\bar{l}} = \frac{\Delta_{\bar{i}j}}{2}\delta_{il}\delta_{jk} - \frac{\Pi_{\bar{i}i,k\bar{k}}}{2}\delta_{ij}\delta_{kl}. \tag{97e}$$

In the $i \neq j$ case, the $\Delta$ sub-matrices with a fixed $i$ and $j$ values must be Hermitian; thence, $\Delta^*_{i\bar{j}} = \Delta_{\bar{j}i}$. From the antisymmetry properties of the 2-RDM, we obtain that $\Delta_{\bar{j}i} = \Delta_{i\bar{j}}$, which makes these matrix elements be real. Thus, the $\Delta$ sub-matrices must be symmetric. The hermiticity of the 2-RDM makes us to impose $\Pi_{\bar{i}i,k\bar{k}} = \Pi^*_{k\bar{k},\bar{i}i}$. And, the antisymmetry of the 2-RDM imposes $\Pi_{\bar{i}i,k\bar{k}} = \Pi_{\bar{i}i,\bar{k}k} = \Pi_{i\bar{i},\bar{k}k} = \Pi_{i\bar{i},k\bar{k}}$.

Using the above definitions for the SD contribution and the cumulant matrix elements, the 2-RDM elements can be approximated as

$$^2D_{ij}^{kl} = \frac{n_i n_j - \Delta_{ij}}{2}\delta_{ik}\delta_{jl} - \frac{n_i n_j - \Delta_{ij}}{2}\delta_{il}\delta_{jk}, \tag{98a}$$

$$^2D_{\bar{i}\bar{j}}^{\bar{k}\bar{l}} = \frac{n_{\bar{i}} n_{\bar{j}} - \Delta_{\bar{i}\bar{j}}}{2}\delta_{ik}\delta_{jl} - \frac{n_{\bar{i}} n_{\bar{j}} - \Delta_{\bar{i}\bar{j}}}{2}\delta_{il}\delta_{jk}, \tag{98b}$$

$$^2D_{i\bar{j}}^{k\bar{l}} = \frac{n_i n_{\bar{j}} - \Delta_{i\bar{j}}}{2}\delta_{ik}\delta_{jl} + \frac{\Pi_{i\bar{i},k\bar{k}}}{2}\delta_{ij}\delta_{kl}, \tag{98c}$$

$$^2D_{\bar{j}i}^{\bar{l}k} = \frac{n_i n_{\bar{j}} - \Delta_{\bar{j}i}}{2}\delta_{ik}\delta_{jl} + \frac{\Pi_{\bar{i}i,\bar{k}k}}{2}\delta_{ij}\delta_{kl}, \tag{98d}$$

$$^2D_{\bar{j}i}^{k\bar{l}} = -\frac{n_i n_{\bar{j}} - \Delta_{\bar{j}i}}{2}\delta_{ik}\delta_{jl} - \frac{\Pi_{\bar{i}i,k\bar{k}}}{2}\delta_{ij}\delta_{kl}, \tag{98e}$$

and

$$^2D_{i\bar{j}}^{\bar{l}k} = -\frac{n_i n_{\bar{j}} - \Delta_{i\bar{j}}}{2}\delta_{ik}\delta_{jl} - \frac{\Pi_{i\bar{i},\bar{k}k}}{2}\delta_{ij}\delta_{kl}. \tag{98f}$$

The antisymmetry properties of the 2-RDM elements implies that Eqs. (98a), (98b), and (98c) provide enough information to build the rest of 2-RDM matrix elements. In contrast to the nonrelativistic approach, the 2-RDM matrix elements given by Eqs. (98e) and (98f) contribute to the energy; we must approximate them in terms of the $\Delta$ and $\Pi$ matrices.

**The D-, Q-, and G- $N$-representability conditions**

The so-called $N$-representability conditions evaluated at the np approximation level are the same as in the nonrelativistic approach; they are associated with the positive semidefinite character of the following Hermitian matrices (see Refs. [75, 86] for more details)

$$D_{IJ}^{KL} = \frac{1}{2}\langle\Psi_+|\widehat{\bar{b}}_I^\dagger\widehat{\bar{b}}_J^\dagger\widehat{\bar{b}}_L\widehat{\bar{b}}_K|\Psi_+\rangle, \tag{99}$$

---

[15]Notice that the $\lambda_{\bar{i}\bar{j},\bar{k}\bar{l}}$ terms are obtained by replacing the unbarred indices by barred ones.

$$Q_{IJ}^{KL} = \frac{1}{2}\langle\Psi_+|\widehat{\widetilde{b}}_I\widehat{\widetilde{b}}_J\widehat{\widetilde{b}}_L^\dagger\widehat{\widetilde{b}}_K^\dagger|\Psi_+\rangle, \tag{100}$$

$$G_{IJ}^{KL} = \frac{1}{2}\langle\Psi_+|\widehat{\widetilde{b}}_I^\dagger\widehat{\widetilde{b}}_J\widehat{\widetilde{b}}_L^\dagger\widehat{\widetilde{b}}_K|\Psi_+\rangle. \tag{101}$$

In the nonrelativistic approach, these matrices can be split in terms of different spin-blocks for the evaluation of the positive semi-definite character.

- The D-Condition:

  Eq. (99) tests the positive semidefinite character of the 2-RDM. From the block structure of the 2-RDM (see Eq. (70)), let us first focus on the block $({}^2\mathbf{D}_{ij}^{ij}, {}^2\mathbf{D}_{ji}^{ij})$, which using Eq. (98a) gives us the condition that for $i \neq j$ there is a set of $[(M/2)\times(M/2-1)]/2$ two-by-two sub-blocks of the form[16]

  $$\begin{pmatrix} \frac{(n_i n_j - \Delta_{ij})}{2} & -\frac{(n_i n_j - \Delta_{ij})}{2} \\ -\frac{(n_i n_j - \Delta_{ij})}{2} & \frac{(n_i n_j - \Delta_{ij})}{2} \end{pmatrix}, \tag{102}$$

  which upon diagonalization produce the eigenvalues 0 and $2(n_i n_j - \Delta_{ij})$. Since the **D** matrix must be positive semidefinite, we can conclude that

  $$\Delta_{ij} \leq n_i n_j. \tag{103}$$

  Similarly, the third block produces the same type of eigenvalues and conditions (i.e. $\Delta_{\bar{i}\bar{j}} \leq n_{\bar{i}} n_{\bar{j}}$). Finally, for the middle block and $i \neq j$ we have a set of $[(M/2)\times(M/2)]$ two-by-two blocks of the form

  $$\begin{pmatrix} \frac{(n_i n_{\bar{j}} - \Delta_{i\bar{j}})}{2} & -\frac{(n_i n_{\bar{j}} - \Delta_{i\bar{j}})}{2} \\ -\frac{(n_i n_{\bar{j}} - \Delta_{i\bar{j}})}{2} & \frac{(n_i n_{\bar{j}} - \Delta_{i\bar{j}})}{2} \end{pmatrix}, \tag{104}$$

  whose eigenvalues are 0 and $2(n_i n_{\bar{j}} - \Delta_{i\bar{j}})$, and to fulfill the D-condition the following constraint must hold

  $$\Delta_{i\bar{j}} \leq n_i n_{\bar{j}}. \tag{105}$$

  Also, for the middle block we have one large $[(M/2)\times(M/2)]$ block whose off-diagonal elements are given by all the $\mathbf{\Pi}$ matrix off-diagonal elements. The diagonal part of this block contains the $i = j$ case, which includes the $\mathbf{\Delta}$ contribution. This large block is basis set dependent and does not lead to analytic expressions of any of the auxiliary matrices; thence, we can not extract any information (constraints) from it.

- The Q-Condition:

  Using the anticommutation rules for the creation and annihilation operators, we may rewrite Eq. (100) as

  $$Q_{IJ}^{KL} = {}^2D_{KL}^{IJ} + \frac{1}{2}\left[\delta_{JL}(\delta_{IK} - {}^1D_K^I) - \delta_{IL}(\delta_{JK} - {}^1D_K^J) + \delta_{JK}{}^1D_L^I - \delta_{IK}{}^1D_L^J\right] \tag{106}$$

  $$= {}^2D_{KL}^{IJ} + \frac{1}{2}\left[\delta_{JL}\delta_{IK}h_K - \delta_{IL}\delta_{JK}h_K + \delta_{JK}\delta_{LI}h_L - \delta_{IK}\delta_{JL}h_L\right], \tag{107}$$

  where we have defined $h_I = 1 - n_I$, and taken the 1-RDM in the NO representation in the last expression. It is straightforward to recognize that the **Q** matrix shows the same

---

[16]For $i = j$, the elements are 0 by definition and they fulfill the condition.

block structure as the $\mathbf{D}$ matrix. Hence, we have one large block that depends on the $\mathbf{\Pi}$ and $\mathbf{\Delta}$ matrices (and on the basis set), and two-by-two blocks for $i \neq j$ of the form

$$
\begin{pmatrix}
\frac{(1-n_i-n_j+n_i n_j-\Delta_{ij})}{2} & -\frac{(1-n_i-n_j+n_i n_j-\Delta_{ij})}{2} \\
-\frac{(1-n_i-n_j+n_i n_j-\Delta_{ij})}{2} & \frac{(1-n_i-n_j+n_i n_j-\Delta_{ij})}{2}
\end{pmatrix}, \tag{108}
$$

$$
\begin{pmatrix}
\frac{(1-n_i-n_{\bar{j}}+n_i n_{\bar{j}}-\Delta_{i\bar{j}})}{2} & -\frac{(1-n_i-n_{\bar{j}}+n_i n_{\bar{j}}-\Delta_{ij})}{2} \\
-\frac{(1-n_i-n_{\bar{j}}+n_i n_{\bar{j}}-\Delta_{ij})}{2} & \frac{(1-n_i-n_{\bar{j}}+n_i n_{\bar{j}}-\Delta_{ij})}{2}
\end{pmatrix}, \tag{109}
$$

that upon diagonalization lead us to the following conditions

$$
\Delta_{ij} \leq h_i h_j \qquad \text{and} \qquad \Delta_{i\bar{j}} \leq h_i h_{\bar{j}}. \tag{110}
$$

- The G-Condition:

As we did for the Q-condition, we may rewrite Eq. (101), using the anticommutation relations of the creation and annihilation operators, as

$$
G_{IJ}^{KL} = -{}^2D_{IL}^{KJ} + \frac{1}{2}\delta_{JL}{}^1D_I^K \tag{111}
$$

$$
= -{}^2D_{IL}^{KJ} + \frac{1}{2}\delta_{JL}\delta_{IK}n_I, \tag{112}
$$

where in the last expression we have used the 1-RDM in the NO representation. For $i \neq j$ the first block $\left({}^2\mathbf{D}_{ij}^{ij}, {}^2\mathbf{D}_{ji}^{ij}\right)$ (and also for the third block) the $\mathbf{G}$ matrix is formed by a large basis set dependent block that does not provide any extra analytical constraint. But it also contains $1 \times 1$ blocks with elements of the form $-{}^2D_{ij}^{ij} + \frac{1}{2}n_i$ that lead to a condition of the form

$$
\Delta_{ij} \geq -h_i n_j, \tag{113}
$$

which is easy to satisfy with ONs between 0 and 1 and $\Delta_{ij} \geq 0$. Using the matrix elements of the central block of the 2-RDM let us introduce the $\mathbf{G}^{\text{central}}$ matrix that reads as

$$
\mathbf{G}^{\text{central}} = \begin{pmatrix}
\left(\mathbf{G}_{\bar{j}i}^{\bar{i}j}, \mathbf{G}_{\bar{i}j}^{\bar{i}j}\Big|_{i\neq j}\right) & \left(\mathbf{G}_{ji}^{\bar{i}j}, \mathbf{G}_{jj}^{\bar{i}i}\Big|_{i\neq j}\right) \\
\left(\mathbf{G}_{\bar{j}i}^{ij}, \mathbf{G}_{\bar{i}i}^{jj}\Big|_{i\neq j}\right) & \left(\mathbf{G}_{ji}^{i\bar{j}}, \mathbf{G}_{ji}^{j\bar{i}}\Big|_{i\neq j}\right)
\end{pmatrix}, \tag{114}
$$

whose elements can be built using Eq. (111). It is easy to recognize that this block can be split into three disjoint sub-blocks (i.e. $\left(\mathbf{G}_{\bar{j}i}^{\bar{i}j}, \mathbf{G}_{\bar{i}j}^{\bar{i}j}\Big|_{i\neq j}\right)$, $\left(\mathbf{G}_{ji}^{i\bar{j}}, \mathbf{G}_{ji}^{j\bar{i}}\Big|_{i\neq j}\right)$, and the rest). The latter sub-block is basis set dependent and does not lead to new information about the auxiliary matrices. On the contrary, the former sub-block provides new information. This block leads to $M/2$ eigenvalues of the form

$$
\frac{-(n_i n_{\bar{i}} - \Delta_{i\bar{i}}) - \Pi_{\bar{i}i,\bar{i}i} + n_{\bar{i}}}{2}, \tag{115}
$$

which fulfill the $N$-representability condition if we let for example $\Pi_{\bar{i}i,\bar{i}i} = n_{\bar{i}}$ and $\Delta_{i\bar{i}} = n_i n_{\bar{i}}$. Nevertheless, this sub-block also produces for $i \neq j$ two-by-two sub-blocks of the form

$$
\begin{pmatrix}
G_{\bar{i}j}^{\bar{i}j} & G_{\bar{j}i}^{\bar{i}j} \\
G_{\bar{i}j}^{\bar{j}i} & G_{\bar{j}i}^{\bar{j}i}
\end{pmatrix} = \frac{1}{2}\begin{pmatrix}
\Delta_{\bar{i}j} + n_{\bar{i}}h_j & -\Pi_{\bar{j}j,\bar{i}i} \\
-\Pi_{\bar{i}i,\bar{j}j} & \Delta_{\bar{j}i} + n_{\bar{j}}h_i
\end{pmatrix}, \tag{116}
$$

whose eigenvalues obtained upon diagonalization read as

$$\frac{1}{4}\left(n_{\bar{i}}h_j + \Delta_{\bar{i}j} + \Delta_{\bar{j}i} + n_{\bar{j}}h_i\right) \pm \frac{1}{4}\sqrt{4\Pi_{\bar{i}i,\bar{j}j}^2 + (n_{\bar{i}}h_j - n_{\bar{j}}h_i)^2}; \tag{117}$$

where we have used the symmetry properties of the $\mathbf{\Pi}$ matrix. The eigenvalue with the positive square root is greater or equal to zero and it satisfies the requirement. On the other hand, the eigenvalue with a negative square root leads to some bounds for the elements of the $\mathbf{\Pi}$ matrix. These bounds depend on the $\mathbf{\Delta}$ matrix and they can be written as

$$\left(n_{\bar{i}}h_j + \Delta_{\bar{i}j} + \Delta_{\bar{j}i} + n_{\bar{j}}h_i\right)^2 \geq 4\Pi_{\bar{i}i,\bar{j}j}^2 + (n_{\bar{i}}h_j - n_{\bar{j}}h_i)^2, \tag{118}$$

$$n_{\bar{i}}h_j(2\Delta_{\bar{i}j} + 2\Delta_{\bar{j}i} + 4n_{\bar{j}}h_i) + \Delta_{\bar{i}j}^2 + \Delta_{\bar{j}i}^2 + 2\Delta_{\bar{j}i}n_{\bar{j}}h_i + \Delta_{\bar{i}j}(2\Delta_{\bar{j}i} + 2n_{\bar{j}}h_i) \geq 4\Pi_{\bar{i}i,\bar{j}j}^2, \tag{119}$$

which in the particular case when $\Delta_{\bar{i}j} = \Delta_{\bar{j}i} = 0$ leads to

$$\sqrt{n_{\bar{i}}h_i n_{\bar{j}}h_j} \geq |\Pi_{\bar{i}i,\bar{j}j}|. \tag{120}$$

Table 3: Definition of the $\mathbf{\Pi}$ matrix elements, where $n_i^d = n_i h_p^d/h_p$ with $i \in \Omega_p$, $h_p = 1 - n_p$, and $h_p^d = h_p \exp\left[-(h_p/(0.02\sqrt{2}))^2\right]$.

| functional | $\Pi_{ij}^{\text{intra}}$; $i,j\in\Omega_p, i\neq j$ | | $\Pi_{ij}^{\text{inter}}$; $i\in\Omega_p, j\in\Omega_q, p\neq q$ | | Ref. |
|---|---|---|---|---|---|
| PNOF5 | $-\sqrt{n_i n_j}$ $\sqrt{n_i n_j}$ | $i=p$ or $j=p$ otherwise | $0$ | all | [120] |
| PNOF7 | $-\sqrt{n_i n_j}$ $\sqrt{n_i n_j}$ | $i=p$ or $j=p$ otherwise | $-\sqrt{n_i n_j h_i h_j}$ | all | [83], [101] |
| PNOF7s | $-\sqrt{n_i n_j}$ $\sqrt{n_i n_j}$ | $i=p$ or $j=p$ otherwise | $-4n_i n_j h_i h_j$ | all | [81] |
| GNOF | $-\sqrt{n_i n_j}$ $\sqrt{n_i n_j}$ | $i=p$ or $j=p$ otherwise | $n_i^d n_j^d - \sqrt{n_i n_j h_i h_j} - \sqrt{n_i^d n_j^d}$ $n_i^d n_j^d - \sqrt{n_i n_j h_i h_j} + \sqrt{n_i^d n_j^d}$ $0$ | $i > p$ and $j = q$ or $i = p$ and $j > q$ $i > p, j > q$ otherwise | [102] |

From this analysis, we conclude that at the np-ReNOFT level the *ansatz* proposed in Eqs. (97a)-(97c) provides equivalent eigenvalues for the D-, Q-, and G-conditions as in the nonrelativistic context. Indeed, only the definition of the auxiliary matrix $\mathbf{\Pi}$ has slightly changed w.r.t. its nonrelativistic counterpart. In principle, in the relativistic approach four indices are required to define this matrix (instead of the two indices used in nonrelativistic NOFT) because now the elements of the form $^2D_{\bar{i}j}^{i\bar{j}}$ contribute to the energy. Nevertheless, when the $\mathbf{\Pi}$ matrix is real and the ONs for barred and unbarred states are the same (i.e. $n_i = n_{\bar{i}}$), only two indices are needed; thus, the Eq. (120) can be rewritten as in the nonrelativistic approach

$$\sqrt{n_i h_i n_j h_j} \geq |\Pi_{i,j}|, \tag{121}$$

with $\Pi_{i,j} = \Pi_{\bar{i}i,\bar{j}j}$.

To ensure that the rel-GNOF/PNOF$x$ functionals used in this work recover their nonrelativistic counterparts in the nonrelativistic limit, we impose the following constraints:

1. Partition the PS space into subspaces $\{\Omega_p\}$ (see Fig. 1 for more details).

2. Let $\Delta_{ij} = \Delta_{i\bar{j}} = \Delta_{\bar{i}j} = \Delta_{\bar{i}\bar{j}}$.

3. Define the diagonal terms $\Pi_{i,i} = n_i$ and $\Delta_{ii} = n_i^2$ to also satisfy Eq. (115).

4. Set $\Delta_{ij} = n_i n_j$ if $i, j \in \Omega_p$ and $\Delta_{ij} = 0$ otherwise (this selection satisfies the above inequalities).

And we arrive to the rel-GNOF/PNOFx functionals presented in this work. The only missing term is the definition of the $\Pi$ matrix elements. In Table 3 we have collected different definitions, which are also employed in the nonrelativistic context; that (as we have seen) are also valid in the relativistic np approximation scenario. Let us remark that the $\Pi$ matrix is separated into two contributions. One contribution formed by indices belonging to the same $\Omega_p$ subspace (named as $\Pi^{\text{intra}}$) that accounts for intra-subspace interactions; the second contribution where the indices belong to different $\Omega$ subspaces (denoted as $\Pi^{\text{inter}}$). Using these terms in the definitions of the 2-RDM elements (Eqs. 98a-98f), we arrive to the approximated (reconstructed) 2-RDM matrix. Inserting this matrix elements in Eq. (64) we arrive to the functionals presented in Eq. (72).

# E   The relativistic Fixed-Phases functional.

In 1956 Löwdin and Shull proved that the configuration interaction coefficients become simple functions of the ONs (i.e. $c_I = \pm\sqrt{n_I}$) when NOs are employed to express the two-electron wavefunction [78]. Thence, in this work, we extend this result to the relativistic case. Let us start our discussion introducing the most general relativistic two-electron wavefunction [17]

$$\Psi_+^{2e-}(\mathbf{r}_1, \mathbf{r}_2) = \sum_{I,J} C_{IJ} \psi_I(\mathbf{r}_1) \otimes \psi_J(\mathbf{r}_2), \tag{122}$$

where the coefficients $C_{IJ} \in \mathbb{C}$ and form an antisymmetric matrix $\mathbf{C}$ (i.e. $C_{IJ} = -C_{JI}$ and $C_{II} = 0$). From the normalization of the wavefunction we have that $\sum_{I,J} |C_{IJ}|^2 = 1$. Applying the Carlson–Keller expansion [121] (a.k.a. the Schmidt decomposition [122]) to the wavefunction (122) we arrive to

$$\Psi_+^{2e-}(\mathbf{r}_1, \mathbf{r}_2) = \frac{1}{\sqrt{2}} \sum_{\tilde{i}} \tilde{c}_{\tilde{i}} \big[ \boldsymbol{\vartheta}_{\tilde{i}}(\mathbf{r}_1) \otimes \boldsymbol{\vartheta}_{\tilde{\tilde{i}}}(\mathbf{r}_2) - \boldsymbol{\vartheta}_{\tilde{i}}(\mathbf{r}_2) \otimes \boldsymbol{\vartheta}_{\tilde{\tilde{i}}}(\mathbf{r}_1) \big], \tag{123}$$

where the index $\tilde{i}$ runs only over half of the PS, and we have introduced the $\boldsymbol{\vartheta}$ spinors that form Schmidt pairs $(\tilde{i}, \tilde{\tilde{i}})$. Note that these Schmidt orbitals are only defined up to a unitary transformation within the degenerate subspace.

The 1-RDM of the wavefunction given by 123 reads as

$$\mathbf{n}_1^{+,2e-}(\mathbf{r}, \mathbf{r}') = \sum_{\tilde{i}} |\tilde{c}_{\tilde{i}}|^2_{\tilde{i}} \Big[ \boldsymbol{\vartheta}_{\tilde{i}}(\mathbf{r}) \boldsymbol{\vartheta}_{\tilde{i}}^\dagger(\mathbf{r}') + \boldsymbol{\vartheta}_{\tilde{\tilde{i}}}(\mathbf{r}) \boldsymbol{\vartheta}_{\tilde{\tilde{i}}}^\dagger(\mathbf{r}') \Big], \tag{124}$$

which is already given in its diagonal representation and from which we find $|\tilde{c}_{\tilde{i}}|^2 = n_{\tilde{i}}$

When the Hamiltonian operator preserves time-reversal symmetry (i.e. it is time-independent and does not contain external magnetic fields) it commutes with the Kramers' operator [18]. This implies that we can choose all our eigenstates to satisfy [123]

$$\Psi_+^{2e-}(\mathbf{r}_1, \mathbf{r}_2) = \widehat{\mathcal{K}} \Psi_+^{2e-}(\mathbf{r}_1, \mathbf{r}_2), \tag{125}$$

---

[17]Recall that the 'spin' information is contained in the 4-component spinors.

[18]$\widehat{\mathcal{K}}$ in the many electron case leads to the complex conjugate of the CI coefficients and transforms all spinors in the tensor product [54, 123]

which has the right symmetry for real CI coefficients. This means that the degenerate configurations in (123) transform into each other. Since the Schmidt orbitals are only defined up to a unitary transformation with the degenerate subspace, we can use this freedom to transform the Schmidt pairs to Kramers pairs. So the Schmidt pairs can be replaced with Kramers pairs

$$\Psi_+^{2e-}(\mathbf{r}_1, \mathbf{r}_2) = \frac{1}{\sqrt{2}} \sum_i c_i \big[ \widetilde{\boldsymbol{\chi}}_i(\mathbf{r}_1) \otimes \widetilde{\boldsymbol{\chi}}_{\bar{i}}(\mathbf{r}_2) - \widetilde{\boldsymbol{\chi}}_i(\mathbf{r}_2) \otimes \widetilde{\boldsymbol{\chi}}_{\bar{i}}(\mathbf{r}_1) \big], \tag{126}$$

where

$$c_i = \sqrt{n_i} e^{-i\zeta_i}. \tag{127}$$

The phases $\zeta_i = k\pi$ with $k \in \mathbb{Z}$ to preserve the Kramers' symmetry.

The 1-RDM is given by

$$\mathbf{n}_1^{+,2e-}(\mathbf{r}, \mathbf{r}') = \sum_i n_i \left[ \widetilde{\boldsymbol{\chi}}_i(\mathbf{r}) \widetilde{\boldsymbol{\chi}}_i^\dagger(\mathbf{r}') + \widetilde{\boldsymbol{\chi}}_{\bar{i}}(\mathbf{r}) \widetilde{\boldsymbol{\chi}}_{\bar{i}}^\dagger(\mathbf{r}') \right]. \tag{128}$$

Evaluating the electron-electron interaction using Eq. (126) we arrive to

$$\langle \Psi_+^{2e-} | \widehat{\widetilde{W}} | \Psi_+^{2e-} \rangle = \sum_{i,j} \frac{\sqrt{n_i n_j}}{2} e^{i(\zeta_j - \zeta_i)} \int d\mathbf{r}_1 d\mathbf{r}_2 \mathrm{Tr} \bigg[ \frac{1}{r_{12}} (\mathbb{I}_{16\times16} - \boldsymbol{\alpha} \odot \boldsymbol{\alpha})$$
$$\times \Big( (\widetilde{\boldsymbol{\chi}}_i(\mathbf{r}_1) \otimes \widetilde{\boldsymbol{\chi}}_{\bar{i}}(\mathbf{r}_2))(\widetilde{\boldsymbol{\chi}}_j^\dagger(\mathbf{r}_1) \otimes \widetilde{\boldsymbol{\chi}}_{\bar{j}}^\dagger(\mathbf{r}_2)) - (\widetilde{\boldsymbol{\chi}}_i(\mathbf{r}_1) \otimes \widetilde{\boldsymbol{\chi}}_{\bar{i}}(\mathbf{r}_2))(\widetilde{\boldsymbol{\chi}}_j^\dagger(\mathbf{r}_2) \otimes \widetilde{\boldsymbol{\chi}}_{\bar{j}}^\dagger(\mathbf{r}_1))$$
$$- (\widetilde{\boldsymbol{\chi}}_i(\mathbf{r}_2) \otimes \widetilde{\boldsymbol{\chi}}_{\bar{i}}(\mathbf{r}_1))(\widetilde{\boldsymbol{\chi}}_j^\dagger(\mathbf{r}_1) \otimes \widetilde{\boldsymbol{\chi}}_{\bar{j}}^\dagger(\mathbf{r}_2)) + (\widetilde{\boldsymbol{\chi}}_i(\mathbf{r}_2) \otimes \widetilde{\boldsymbol{\chi}}_{\bar{i}}(\mathbf{r}_1))(\widetilde{\boldsymbol{\chi}}_j^\dagger(\mathbf{r}_2) \otimes \widetilde{\boldsymbol{\chi}}_{\bar{j}}^\dagger(\mathbf{r}_1)) \Big) \bigg], \tag{129}$$

where $\mathbb{I}_{16\times16}$ is the 16×16 identity matrix. In this form, we notice that the energy minimization procedure also implies the optimization w.r.t. the phases ($\{\zeta_i\}$). Consequently, fixing the phases as in the nonrelativistic case and exploding Kramers' symmetry on the integrals we arrive to the energy expression given by Eq. (77). Therefore, the rel-GNOF/PNOF$x$ functionals are equivalent to the relativistic version of the Fixed-Phases functional for 'spin-compensated' two-electron systems.

Finally, let us show that the relativistic Fixed-Phases functional contains its nonrelativistic counterpart. To that end, let us first focus on the NOs forming the Kramers' pairs, which are built as four component spinors, i.e.

$$\widetilde{\boldsymbol{\chi}}_i(\mathbf{r}) = \begin{pmatrix} \widetilde{\phi}_{i,1}(\mathbf{r}) \\ \widetilde{\phi}_{i,2}(\mathbf{r}) \\ \widetilde{\phi}_{i,3}(\mathbf{r}) \\ \widetilde{\phi}_{i,4}(\mathbf{r}) \end{pmatrix} \tag{130}$$

and

$$\widehat{\mathcal{K}} \widetilde{\boldsymbol{\chi}}_i(\mathbf{r}) = \begin{pmatrix} -\widetilde{\phi}_{i,2}^*(\mathbf{r}) \\ \widetilde{\phi}_{i,1}^*(\mathbf{r}) \\ -\widetilde{\phi}_{i,4}^*(\mathbf{r}) \\ \widetilde{\phi}_{i,3}^*(\mathbf{r}) \end{pmatrix} = \widetilde{\boldsymbol{\chi}}_{\bar{i}}(\mathbf{r}). \tag{131}$$

It is easy to prove that the tensor product of 4-component NOs contains singlet and triplet contributions, but we can introduce some constraints to remove the triplet contribution. To that end, let us assume that the so-called small component terms of the NOs are negligible (i.e. $\widetilde{\phi}_{i,3}(\mathbf{r})$ and $\widetilde{\phi}_{i,4}(\mathbf{r}) \rightarrow 0$) [19]. Then, adding

---

[19]This is normally the case in the nonrelativistic limit [54]

the following constraint $\widetilde{\phi}_{i,2}(\mathbf{r}) = a\widetilde{\phi}_{i,1}(\mathbf{r})$ with $a \in \mathbb{C}$ we may write the components as $\widetilde{\phi}_{i,1}(\mathbf{r}) = \Re\widetilde{\phi}_{i,1}(\mathbf{r}) + \mathrm{i}\Im\widetilde{\phi}_{i,1}(\mathbf{r})$ and $\widetilde{\phi}_{i,2}(\mathbf{r}) = a\Re\widetilde{\phi}_{i,1}(\mathbf{r}) + a\mathrm{i}\Im\widetilde{\phi}_{i,1}(\mathbf{r})$, where $\Re\widetilde{\phi}_{i,1}(\mathbf{r}) = \mathrm{Real}\left[\widetilde{\phi}_{i,1}(\mathbf{r})\right]$ and $\Im\widetilde{\phi}_{i,1}(\mathbf{r}) = \mathrm{Imaginary}\left[\widetilde{\phi}_{i,1}(\mathbf{r})\right]$. Thence, we arrive to the following expression for the NOs

$$\widetilde{\chi}_i(\mathbf{r}) = \begin{pmatrix} \Re\widetilde{\phi}_{i,1}(\mathbf{r}) + \mathrm{i}\Im\widetilde{\phi}_{i,1}(\mathbf{r}) \\ a\Re\widetilde{\phi}_{i,1}(\mathbf{r}) + a\mathrm{i}\Im\widetilde{\phi}_{i,1}(\mathbf{r}) \end{pmatrix} \tag{132}$$

and

$$\widetilde{\chi}_{\bar{i}}(\mathbf{r}) = \begin{pmatrix} -a^*\Re\widetilde{\phi}_{i,1}(\mathbf{r}) + a^*\mathrm{i}\Im\widetilde{\phi}_{i,1}(\mathbf{r}) \\ \Re\widetilde{\phi}_{i,1}(\mathbf{r}) - \mathrm{i}\Im\widetilde{\phi}_{i,1}(\mathbf{r}) \end{pmatrix}, \tag{133}$$

where $a^*$ is the complex-conjugate of $a$ and we have already omitted the small component terms.

Assuming that the scalar spinors are real (i.e. $\Im\widetilde{\phi}_{i,1}(\mathbf{r}) = 0$) and inserting the NOs in Eq. (126) we arrive to [20]

$$\Psi_+^{2e-}(\mathbf{r}_1,\mathbf{r}_2) = \frac{1}{\sqrt{2}}\sum_i c_i\left[\Re\widetilde{\phi}_{i,1}(\mathbf{r}_1)\begin{pmatrix}1\\a\end{pmatrix}\Re\widetilde{\phi}_{i,1}(\mathbf{r}_2)\begin{pmatrix}-a^*\\1\end{pmatrix} - \Re\widetilde{\phi}_{i,1}(\mathbf{r}_2)\begin{pmatrix}1\\a\end{pmatrix}\Re\widetilde{\phi}_{i,1}(\mathbf{r}_1)\begin{pmatrix}-a^*\\1\end{pmatrix}\right],$$
$$\tag{134}$$

which in the $a = 0$ case allows us to introduce the usual nonrelativistic spin functions $\alpha = \begin{pmatrix}1\\0\end{pmatrix}$ and $\beta = \begin{pmatrix}0\\1\end{pmatrix}$ [21]. Inserting the spin functions may rewrite the wavefunction in a simplified notation as

$$\Psi_+^{2e-}(\mathbf{r}_1,\mathbf{r}_2) = \frac{1}{\sqrt{2}}\sum_i c_i \Re\widetilde{\phi}_{i,1}(\mathbf{r}_1)\Re\widetilde{\phi}_{i,1}(\mathbf{r}_2)(\alpha\beta - \beta\alpha) \tag{135}$$

that corresponds to the singlet wavefunction in the nonrelativistic limit with the $\{\Re\widetilde{\phi}_{i,1}\}$ being the scalar NOs. Indeed, from Eq. (135) the nonrelativistic Fixed-Phases functional can be readily obtained; therefore, the relativistic Fixed-Phases functional presented in this work contains its nonrelativistic counterpart.

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
