# Peer review of "Relativistic reduced density matrix functional theory."

_SciPost Chemistry, doi:SciPost Chem. 1, 004 (2022)_

## Round 1 · Referee Report · Julien Toulouse (Referee 1) · 2022-3-10

Strengths

1- The paper extends reduced density-matrix functional theory to effective QED, providing a potentially new powerful method for accurate relativistic many-body calculations. 2- The theory is comprehensively and well explained, with a lot of appendices providing useful proofs and explanations. 3- Different levels of practical approximations are proposed.

Weaknesses

1- The issue with QED renormalization is not mentioned. Even though this goes beyond the scope of the present work, it would be nice to at least mention that an implementation of the present theory based on effective QED would need to address this issue. 2- The justification for using non-relativistic RDMFT functionals is not clear. In relativistic DFT, one should in principle develop new exchange-correlation density functionals including relativistic effects. In RDMFT, since we need only a functional for the two-body interaction (and not for the kinetic-energy operator) which has essentially the same form as in the non-relativistic theory, does this mean that, at least in the no-pair approximation, the exact relativistic exchange-correlation density-matrix functional has an identical expression to the non-relativistic one? This point could be clarified.

Report

This is an excellent paper that represents a very significant advance in the development of sound reduced density-matrix functional theories for many-body relativistic calculations. I recommend publication in SciPost Chem, after that the minor points listed below are addressed.

Requested changes

Section I: 1- The sentence "the popular working horse of physicists and chemists (i.e. the use of Kohn–Sham DFT approach), in general, fails to account for nondynamic correlation effects" may be misleading since Kohn–Sham DFT is an exact approach. It would be clearer to write "Kohn–Sham DFT approach with the usual density-functional approximations". 2- Replacing "Iracane and coworkers" by "Chaix et al." would be more conform to citation conventions.

Section II.A: 3- In Eqs. (1) and (2), the notation "alpha_r" is weird. I think it should be just "alpha".

Section II.B: 4- "hermitian" should be spelled "Hermitian". 5- In Eqs. (9) and (10), why having v_ext(r',r) instead of v_ext(r,r')? In particular, it would seem more conventional to have "\int dr' v_ext(r,r') \psi_A(r')" in Eq. (10). 6- The "charge operator" defined in Eq. (13) is actually the "opposite charge operator". This should be mentioned to avoid confusion. 7- In the paragraph after Eq. (13), the Hamiltonian \tilde{H}_0^v is introduced without an explicit definition. I understand that this Hamiltonian is defined like in Eqs. (11) and (12) but using now the normal ordering \tilde{N} with respect to the new basis. I would advice to give an explicit definition to avoid confusion. In particular, if one forgets about normal ordering, one may be confused about why changing basis should lead to a different Hamiltonian. Also, this would avoid confusion about whether the vacuum energy is included or not. 8- In Eq. (21), I don't understand why the Dirac field operator on the left-hand-side of the equation has a tilde. I think the Dirac field operator should be independent on the basis used (since the orbital rotation matrix cancels out between a_A and \psi_A(r)).

Section II.C: 9- I don't understand why the Hamiltonian \tilde{H}_0^v is subtracted in the definition of the vacuum energy in Eq. (24). I understand that its contribution is zero anyway, but why to include it in the first place? Same question for why the operator \tilde{n}_1 is subtracted in the definition of the vacuum polarization density matrix in Eq. (26). 10- The passage "Before proceeding, let us remark that vp effects are present whenever spinor rotations mixing the positive and negative spinors are involved and the reference effective vacuum changes.[...] VP effects are then present and cannot be avoided. The only way to avoid vp effects is to employ a configuration interaction (CI) vacuum state..." may be confusing because "VP effects" may be understood as physical effects that are still present when doing a CI calculation. Maybe the authors should instead talk about "spinor or orbital relaxation due to VP effects".

Section II.D: 11- Paragraph after Eq. (33), same as the previous remark about "VP effects". Moreover, the authors remain vague about the possibility of defining a generalized normal ordering with respect to the CI vacuum state. Could they say more about the potential difficulties?

Section II.F: 12- In Eqs. (41) and (42), the notations "W^n[n]" and "D_Q^n" are confusing because "^n" is used as a label but may be mistaken for a dependence on the density function "n". I would suggest instead "F[n]" and perhaps "D_Q^dens".

Section III.A: 13- Last paragraph, same as the previous remarks about "VP effects" in Sections II.C and II.D.

Section III.B: 14- First sentence after Eq. (52), I think it should be "<Psi_+ | W | Psi_+>", i.e. without the tilde on W. Indeed, this is the W without tilde which gives rise to the VP contributions of Eqs. (44) and (45).

Section III.C: 15- In Eq. (53), the authors use the Hamiltonian "\tilde{H}[n_1^+]". I understand that they want to emphasize that the Hamiltonian \tilde{H} depends on the orbitals through the normal ordering but I fail to see why it should depend on the orbitals only through the density matrix n_1^+. What is clear is that the expectation value <Psi_+|\tilde{H}|Psi_+> is a functional of only n_1^+ and n_2^+, and thus is a functional of Psi_+, so that the minimization in Eq. (53) makes sense. So I think that the "[n_1^+]" should be dropped. Besides, having no dependence on n_1^+ for \tilde{H} is consistent with Eq. (54) where no such dependence on n_1^+ is indicated for \tilde{W}. 16- For Eq. (57), Ref. 72 is cited, but it should be Ref. 42 instead.

Section III.D: 16- The interaction in Eq. (60) is the Coulomb-Gaunt interaction, and not simply the magnetic Gaunt interaction.

Section III.E: 17- Again, the notation "alpha_r" and "alpha_r'" in Eqs. (62) and (63c) is weird since alpha does not depend on r. Strictly speaking, the product of the two alpha matrices should be the combination of a tensor product ⊗ and a scalar product . , so maybe one could use a symbol such as ⊙ ? Alternatively, one could simply use "alpha_1 . alpha_2" to mean that alpha_1 applies to the first electron and alpha_2 to the second electron. The same remark applies to Eq. (E8). 18- In Eq. (65), the lower label "SD" on the left of D^2 is weird. It is not consistent with the notation in Eq. (71). Also, the lower left label "x" in Eq. (68) (and in the paragraph below) is not explained and does not seem useful. 19- After Eq. (68), the symbol "V_ee" is not defined. 20- In Eq. (76), it seems that \Pi_{i,j}^inter depends on the indices a and b but this is not reflected in the notation.

Section III.F: 21- In Eq. (77), the notation "2e-" should be explained. 22- In Eq. (78) and following equations, the number of electrons is denoted by "N" while it was "N_e" in the previous sections. 23- In Eqs. (79), (80), (82), (83), the notation "i \in a" seems weird since both i and a are indices, and it is not explained.

Appendix A: 24- In Eq. (A1), there is a typo with an index "R". 25- Could the authors extend their proof to the commutation of the charge operator with the two-body interaction operator W?

Appendix C: 26- In Eq. (C1), there are some n_1^vp with tilde and some others without tilde.

Appendix E: 27- The notation "Re" and "Im" as lower indices is weird.

  • validity: top
  • significance: top
  • originality: high
  • clarity: high
  • formatting: excellent
  • grammar: perfect

Author:  Mauricio Rodríguez-Mayorga  on 2022-04-25  [id 2419]

(in reply to Report 1 by Julien Toulouse on 2022-03-10)
Category:
remark
question

First of all, we would like to thank Prof. J. Toulouse for his comments and suggestions to improve our work. In the following, we provide a response to all points requested and include the changes performed on the manuscript.

Weakness

1- The issue with QED renormalization is not mentioned. Even though this goes beyond the scope of the present work, it would be nice to at least mention that an implementation of the present theory based on effective QED would need to address this issue.

We agree with Prof. Toulouse that this point needs to be mentioned. Thus, we have included a paragraph to indicate this issue (including the corresponding references).

2- The justification for using non-relativistic RDMFT functionals is not clear.
In relativistic DFT, one should in principle develop new exchange-correlation density functionals including relativistic effects. In RDMFT, since we need only a functional for the two-body interaction (and not for the kinetic-energy operator) which has essentially the same form as in the non-relativistic theory, does this mean that, at least in the no-pair approximation, the exact relativistic exchange-correlation density-matrix functional has an identical expression to the non-relativistic one? This point could be clarified.

We agree with Prof. Toulouse that this point was not clear enough and we have added a paragraph at the end of the np-ReNOFT functional approximations about imposing Kramers’ symmetry. We hope that now the justification is more clear.

Section I:
1- The sentence "the popular working horse of physicists and chemists (i.e. the use of Kohn–Sham DFT approach), in general, fails to account for nondynamic correlation effects" may be misleading since Kohn–Sham DFT is an exact approach. It would be clearer to write "Kohn–Sham DFT approach with the usual density-functional approximations".

We agree with the referee and we have included the change suggested.

**
2- Replacing "Iracane and coworkers" by "Chaix et al." would be more conform to citation conventions.
**

We have updated the manuscript including this change.

**Section II.A:
3- In Eqs. (1) and (2), the notation "alpha_r" is weird. I think it should be just "alpha".
**

We agree with the referee and have included this change in the updated document. 
 Section II.B:
4- "hermitian" should be spelled "Hermitian".

We thank the referee for noticing this mispelling that we have corrected in the updated manuscript.

**
5- In Eqs. (9) and (10), why having v_ext(r',r) instead of v_ext(r,r')? In particular, it would seem more conventional to have "\int dr' v_ext(r,r') \psi_A(r')" in Eq. (10).** 
 We agree with Prof. Toulouse and we have updated Eqs. 9 and 10 accordingly.   

6- The "charge operator" defined in Eq. (13) is actually the "opposite charge operator". This should be mentioned to avoid confusion.

We thank Prof. Toulouse for pointing this out and we have updated the manuscript following his advice.

**
7- In the paragraph after Eq. (13), the Hamiltonian \tilde{H}_0^v is introduced without an explicit definition. I understand that this Hamiltonian is defined like in Eqs. (11) and (12) but using now the normal ordering \tilde{N} with respect to the new basis. I would advice to give an explicit definition to avoid confusion. In particular, if one forgets about normal ordering, one may be confused about why changing basis should lead to a different Hamiltonian. Also, this would avoid confusion about whether the vacuum energy is included or not.** 
 We thank the referee for pointing this out. We have rephrased the definition to avoid any confusion.

**8- In Eq. (21), I don't understand why the Dirac field operator on the left-hand-side of the equation has a tilde. I think the Dirac field operator should be independent on the basis used (since the orbital rotation matrix cancels out between a_A and \psi_A(r)).
**

We apologize for this error and we have updated the manuscript including this change.

**
Section II.C:
9- I don't understand why the Hamiltonian \tilde{H}_0^v is subtracted in the definition of the vacuum energy in Eq. (24). I understand that its contribution is zero anyway, but why to include it in the first place? Same question for why the operator \tilde{n}_1 is subtracted in the definition of the vacuum polarization density matrix in Eq. (26).**

We were trying to point out the origin of the vp 1-RDM, but we agree that it was more confusing so we have rearranged these paragraphs and removed the zero contributions.

10- The passage "Before proceeding, let us remark that vp effects are present whenever spinor rotations mixing the positive and negative spinors are involved and the reference effective vacuum changes.[...] VP effects are then present and cannot be avoided. The only way to avoid vp effects is to employ a configuration interaction (CI) vacuum state..." may be confusing because "VP effects" may be understood as physical effects that are still present when doing a CI calculation. Maybe the authors should instead talk about "spinor or orbital relaxation due to VP effects".

We agree with the referee comment and we have modified this paragraph as suggested.

Section II.D:
11- Paragraph after Eq. (33), same as the previous remark about "VP effects". Moreover, the authors remain vague about the possibility of defining a generalized normal ordering with respect to the CI vacuum state. Could they say more about the potential difficulties?

We thank the referee for this comment and we have improved the paragraph. We have also added later a comment on the potential difficulties when a CI vacuum is employed.

Section II.F:
12- In Eqs. (41) and (42), the notations "W^n[n]" and "D_Q^n" are confusing because "^n" is used as a label but may be mistaken for a dependence on the density function "n". I would suggest instead "F[n]" and perhaps "D_Q^dens". 
 We agree that this notation may leads to confusion. We would like to thank Prof. Toulouse for the notation suggested that we have employed in the updated the manuscript.   

**
Section III.A:
13- Last paragraph, same as the previous remarks about "VP effects" in Sections II.C and II.D.
**

We thank the referee for this comment and we have improved the paragraph. We have also added later a comment on the potential difficulties when a CI vacuum is employed.

**Section III.B:
14- First sentence after Eq. (52), I think it should be "<Psi_+ | W | Psi_+>", i.e. without the tilde on W. Indeed, this is the W without tilde which gives rise to the VP contributions of Eqs. (44) and (45).
**

We thank Prof. Toulouse for noticing this error that we have corrected in the updated manuscript.

**Section III.C:
15- In Eq. (53), the authors use the Hamiltonian "\tilde{H}[n_1^+]". I understand that they want to emphasize that the Hamiltonian \tilde{H} depends on the orbitals through the normal ordering but I fail to see why it should depend on the orbitals only through the density matrix n_1^+. What is clear is that the expectation value <Psi_+|\tilde{H}|Psi_+> is a functional of only n_1^+ and n_2^+, and thus is a functional of Psi_+, so that the minimization in Eq. (53) makes sense. So I think that the "[n_1^+]" should be dropped. Besides, having no dependence on n_1^+ for \tilde{H} is consistent with Eq. (54) where no such dependence on n_1^+ is indicated for \tilde{W}.
**

We agree with Prof. Toulouse that this notation leads to confusion so we have dropped it in the updated manuscript.

**16- For Eq. (57), Ref. 72 is cited, but it should be Ref. 42 instead.
**

We thank Prof. Toulouse for noticing this mistake that we have corrected in the latest version of the manuscript.

**Section III.D:
16- The interaction in Eq. (60) is the Coulomb-Gaunt interaction, and not simply the magnetic Gaunt interaction.
**

We thank the referee for pointing this out and we have corrected this.

Section III.E:
17- Again, the notation "alpha_r" and "alpha_r'" in Eqs. (62) and (63c) is weird since alpha does not depend on r. Strictly speaking, the product of the two alpha matrices should be the combination of a tensor product ⊗ and a scalar product . , so maybe one could use a symbol such as ⊙ ? Alternatively, one could simply use "alpha_1 . alpha_2" to mean that alpha_1 applies to the first electron and alpha_2 to the second electron. The same remark applies to Eq. (E8).

We would like to thank Prof. Toulouse for suggesting this change on the notation. We use ⊙ in the updated manuscript.

**18- In Eq. (65), the lower label "SD" on the left of D^2 is weird. It is not consistent with the notation in Eq. (71). Also, the lower left label "x" in Eq. (68) (and in the paragraph below) is not explained and does not seem useful.
**

We agree that the notation was not clear and we have addressed this issue in the manuscript.

19- After Eq. (68), the symbol "V_ee" is not defined.

The symbol V_ee was not needed so we have replaced this by writing out electron-electron interaction energy in the manuscript.

**20- In Eq. (76), it seems that \Pi_{i,j}^inter depends on the indices a and b but this is not reflected in the notation.
**

We agree with Prof. Toulouse and we have reflected in the notation the dependence w.r.t. to the new subspaces indices p and q.    Indeed, in rel-PNOF5 and rel-PNOF7 there is no dependence, and only for rel-GNOF it was missing.

**Section III.F:
21- In Eq. (77), the notation "2e-" should be explained.
**

We thank the referee for this suggestion and we have explain the 2e- label appropriately.

**22- In Eq. (78) and following equations, the number of electrons is denoted by "N" while it was "N_e" in the previous sections.
**

We thank the referee for noticing this inconsistency that we have solved on the updated manuscript.

23- In Eqs. (79), (80), (82), (83), the notation "i \in a" seems weird since both i and a are indices, and it is not explained.

This notation was taken from a previous work, where the relationship between PNOF5 and APSG was established. Nevertheless, we agree that it might be problematic for the reader so we have modified it using p and q for electrons pairs (that define different orbital subspaces). And, including subspaces of spinors on the bounds on the sums.   

Appendix A:
24- In Eq. (A1), there is a typo with an index "R". 
 We thank the referee for noticing this typo that we have corrected in the new version.

**25- Could the authors extend their proof to the commutation of the charge operator with the two-body interaction operator W?
**

We thank the referee for this suggestion, but it is something we plan to investigate in the near future.

**Appendix C:
26- In Eq. (C1), there are some n_1^vp with tilde and some others without tilde.
**

We apologize for this error that we have corrected in the updated manuscript.

**Appendix E:
27- The notation "Re" and "Im" as lower indices is weird.   **    

We agree with the referee and we have modified the document with a more usual notation.

---

## Round 1 · Referee Report · Anonymous (Referee 2) · 2022-3-18

Report

In this work presents the theoretical foundations for the relativistic reduced density matrix functional theory. The authors introduced ReRDMFT at three different levels of theory: the full ReRDMFT, two no-pair approximate versions: npvp- and np-ReRDMFT. In particular, total energy can be expressed as the functional of 2-RDM. This is interesting as it connects to nonrelativistic RDMFT, and some approximations to 2-RDM, such as BB, Power, and PNOF functionals, can be used in relativistic RDMFT. The work is very interesting. I would recommend the publication of this work if the author addressed the following issues properly.
1) The authors stated DFT ‘in general, fails to account for nondynamic correlation effects’. Actually, some functionals developed recently can provide good performance for strong correlation, such as the Fermi–Lowdin orbital self-interaction correction (FLOSIC), the fractional spin correction (FSLOSC), and Becke’s B13 functional, etc. It will be helpful for the readers if the authors clarify that explicitly
2) Appendix B extends the Gilbert’s theorem to the relativistic domain. It would be better if the authors stated that the two nonlocal external potentials are different more than a constant, otherwise, their ground states would be the same.
  • validity: high
  • significance: high
  • originality: high
  • clarity: high
  • formatting: excellent
  • grammar: excellent

Author:  Mauricio Rodríguez-Mayorga  on 2022-04-25  [id 2418]

(in reply to Report 2 on 2022-03-18)
Category:
remark
question

First of all, we would like to thank the referee for the comments and suggestions proposed to improve our work.

1) The authors stated DFT ‘in general, fails to account for nondynamic correlation effects’. Actually, some functionals developed recently can provide good performance for strong correlation, such as the Fermi–Lowdin orbital self-interaction correction (FLOSIC), the fractional spin correction (FSLOSC), and Becke’s B13 functional, etc. It will be helpful for the readers if the authors clarify that explicitly.**

This is related to one of the points of the first reviewer, who also objected to this sentence in the Introduction. We have changed this sentence and included the above-mentioned references in this revision.

2) Appendix B extends the Gilbert’s theorem to the relativistic domain. It would be better if the authors stated that the two nonlocal external potentials are different more than a constant, otherwise, their ground states would be the same.

We thank the referee for this suggestion that we have adopted in the final version of our manuscript.

---

## Round 1 · Referee Report · Anonymous (Referee 3) · 2022-3-30

Strengths

1) Theoretical foundation of relativistic RDMFT, with possible applications to a wide set of systems and properties. 2) Clear derivation supported by appendices for details

Weaknesses

  • approximations to the electron-electron interaction: not clear if they are a simple extension of non relativistic approximations (for example do they reduce to the non-relativistic ones using only the Coulomb interaction?) or if they are relativistic approximations.
  • cost of this new framework are not mentioned. the total energy should be optimised with respect to natural orbitals, which are spinors, and occupation numbers. Usually the optimisation with respect to natural orbitals is heavy. Do the authors think that using spinors will complicate the optimisation?

Report

In this work the authors set the foundation of a relativistic version of RDMFT. The authors proceed along three main step: 1) starting from the (free as well as interacting) Dirac equation a relativistic version of RDMFT is set; 2) the no pair approximation is introduced, which allows one to concentrate only on positive energy solutions (electrons); 3) approximations to the electron-electron interaction in terms of occupation numbers are proposed.

The subject of the manuscript is not easy to treat, but the authors made an effort to make it accessible, notably with several appendices where more details are given. I find that this work is interesting and that it might enlarge the field of applicability of RDMFT. In my opinion it can be considered for publication in SciPost Chemistry after some issues reported below are addressed.

Requested changes

1) the authors mentioned that DFT in general fails to account for non dynamic correlation effects, but I don’t agree. First, are the the current approximations used that fail; second, it depends on the property on looks at. For example, if one wants to use the KS density matrix or the KS entropy it is clear that, even with an exact functional, the results are wrong, because the KS system is only supposed to deliver the density of the interacting system. The author should maybe clarify this point.

2) Along the same line: since the one-body density matrix is the key quantity of RDMFT, it is clear that one can get more easily than DFT fractional occupation numbers and better total energies at dissociation. However, I feel that also in RDMFT, as in DFT, one faces the problem that other ground state properties cannot easily be accessed. Maybe the author can comment on this in the introduction.

3) I don’t understand Eq. 24. If I am not mistaken the effective vacuum of \tilde{H} is zero, but then it is used in the definition of \tilde{n}_1^{vp} on the right-hand side. What is this important?

4) The link between the density matrix n_1 and the current density at page 8 is interesting. In relativistic DFT one works with the 4-component current, which scalar part is the standard density and the vector part the standard (3-component) current density. n_1 is also a 4-component quantity, I guess. The scalar part should be the standard one-body density matrix. What is the vector part?

5) The introduction of the vacuum polarization contribution in Eq 43 and 44 is not clear to me. Maybe the authors could try to make it clearer.

6) Why are the vp contributions to the total energy usually neglected in the no pair approximation?

7) Is there a particular reason to use only the Coulomb-Gaut interaction for the formulation of the no-pair relativistic functional approximations?

8) After Eq. (67) the authors state that the two paths to build ReNOFT that they use approximate the 2-RDM elements as functions of the occupation numbers and “ignore the effect of the spinors”. What does that mean? That the natural orbitals are frozen? If this is the case, why? In non-relativistic RDMFT in principle the 2-RDM approximations are functionals of both natural orbitals and occupation numbers.

9) It is not completely clear to me if the approximations to the 2-RDM are non relativistic approximations used in a relativistic context or relativistic approximation which are built following similar steps as in the non relativistic case. For example, is one uses only the Coulomb interaction does one retrieve the same expression as in nonrelativistic RDMFT? It doesn’t seem so, since there are the terms L in the total energy. Maybe the authors can clarify this point.

10) Related to point 8: could the author estimate the cost of the practical application of the theory?

11) some typos, such as: "settling the theoretical foundation"--->"setting the theoretical foundation" "with sets the zero of the energy scale"--->"which sets the zero of the energy scale" "it possible to distinguish"--->"it is possible to distinguish" "only up two indices"--->"only up to two indices" "a equivalent"---> "an equivalent"

  • validity: high
  • significance: good
  • originality: good
  • clarity: high
  • formatting: excellent
  • grammar: excellent

Author:  Mauricio Rodríguez-Mayorga  on 2022-04-25  [id 2417]

(in reply to Report 3 on 2022-03-30)
Category:
question

First of all, we would like to thank the referee for the comments and suggestions proposed to improve our work. Weaknesses - approximations to the electron-electron interaction: not clear if they are a simple extension of non relativistic approximations (for example do they reduce to the non-relativistic ones using only the Coulomb interaction?) or if they are relativistic approximations.

We agree with the referee and we have added a paragraph in section E before introducing the functional approximations to clarify this point.

- cost of this new framework are not mentioned. the total energy should be optimised with respect to natural orbitals, which are spinors, and occupation numbers. Usually the optimisation with respect to natural orbitals is heavy. Do the authors think that using spinors will complicate the optimisation?

We thank the referee for pointing this out and decided to include a paragraph in the conclusions and final remarks section to briefly comment on this issue.

Report In this work the authors set the foundation of a relativistic version of RDMFT. The authors proceed along three main step: 1) starting from the (free as well as interacting) Dirac equation a relativistic version of RDMFT is set; 2) the no pair approximation is introduced, which allows one to concentrate only on positive energy solutions (electrons); 3) approximations to the electron-electron interaction in terms of occupation numbers are proposed. The subject of the manuscript is not easy to treat, but the authors made an effort to make it accessible, notably with several appendices where more details are given. I find that this work is interesting and that it might enlarge the field of applicability of RDMFT. In my opinion it can be considered for publication in SciPost Chemistry after some issues reported below are addressed.

Requested changes 1) the authors mentioned that DFT in general fails to account for non dynamic correlation effects, but I don’t agree. First, are the the current approximations used that fail; second, it depends on the property on looks at. For example, if one wants to use the KS density matrix or the KS entropy it is clear that, even with an exact functional, the results are wrong, because the KS system is only supposed to deliver the density of the interacting system. The author should maybe clarify this point.

We agree with this referee and the other two reviewers, who also objected to this sentence in the Introduction, and have revised this sentence to make clear the problem is in the density functional approximations that are used rather than in density functional theory itself. 

2) Along the same line: since the one-body density matrix is the key quantity of RDMFT, it is clear that one can get more easily than DFT fractional occupation numbers and better total energies at dissociation. However, I feel that also in RDMFT, as in DFT, one faces the problem that other ground state properties cannot easily be accessed. Maybe the author can comment on this in the introduction.

We thank the referee for this comment, but like to stress that an advantage of RDMFT over DFT lies in the fact that all properties which can be expressed with one-body operators can be computed from the 1-RDM (e.g. interaction with magnetic fields, dipole moments, etc.). Moreover, the functional approximations presented in this work allow us to reconstruct 2-RDM matrix elements, which then provides access also to two-body quantities (like pair-probability densities, delocalization indices, intracule probability densities, etc.) as well. While Kohn-Sham DFT employs a auxiliary system to reproduce the actual density, in RDMFT the actual 1-RDM of the system (for a given functional approximation) is obtained without the need of any auxiliary system. 

3) I don’t understand Eq. 24. If I am not mistaken the effective vacuum of \tilde{H} is zero, but then it is used in the definition of \tilde{n}_1^{vp} on the right-hand side. What is this important?

We agree with the referee that including this zero term can be confusing and have modified this section.

4) The link between the density matrix n_1 and the current density at page 8 is interesting. In relativistic DFT one works with the 4-component current, which scalar part is the standard density and the vector part the standard (3-component) current density. n_1 is also a 4-component quantity, I guess. The scalar part should be the standard one-body density matrix. What is the vector part?

We thank the referee for this question. Actually, the 1-RDM is a matrix quantity, so there are 4x4=16 different spinor blocks (only 10 of them unique due to hermiticity). The trace (scalar part) corresponds to the the usual one-body density matrix as it was pointed out by the referee. One could then use the traceless diagonal part to define a vector quantity but that would not be very useful in our opinion, since this would be a vector in spinor space and not in coordinate space. Only after multiplication by the three alpha matrices (alpha_x,alpha_y,alpha_z) it acquires the usual vectorial character of the current. That is to say, by itself the 1-RDM does not contain a vector part, it is acquired upon multiplication by the alpha matrices. We hope this explanation clarifies this point.

5) The introduction of the vacuum polarization contribution in Eq 43 and 44 is not clear to me. Maybe the authors could try to make it clearer.

We thank the referee for this suggestion. We hope that the modifications already performed in the Vacuum polarization section (Eqs. 24-27) clarify the introduction of vacuum polarization effects (and energies) and facilitate the introduction of VP effects also in the fermion-fermion interacting case. 

6) Why are the vp contributions to the total energy usually neglected in the no pair approximation?

We thank the referee for this question. From the theoretical perspective it forces the method to deal with IR and UV divergences, which requires a proper renormalization scheme (as we have commented upon in the updated manuscript). From the computational/practical point of view, it requires computing Eqs. 43 and 44, which will increase the computational cost as more repulsion integrals need to be evaluated explicitly (e.g. Hartree and exchange integrals between the occupied and the NS enter the evaluation of the hat{tilde{V}}^VP contribution). In addition this puts higher demands on the finite basis set that is used as it should also provide a correct representation of the negative energy solutions.

7) Is there a particular reason to use only the Coulomb-Gaut interaction for the formulation of the no-pair relativistic functional approximations?

We thank the referee for this question. Despite the theoretical foundations of ReRDMFT allow us to use other interactions (e.g. Coulomb-Breit interaction), from the practical perspective it is common to retain/employ only the Coulomb-Gaunt interaction. This is because the electron repulsion integrals are easy to compute as they be written as linear combinations of integrals evaluated in a standard nonrelativistic integral program. Obtaining integrals over the full Breit operator requires integrals that are usually only evaluated for higher order geometrical derivatives.    

** 

8) After Eq. (67) the authors state that the two paths to build ReNOFT that they use approximate the 2-RDM elements as functions of the occupation numbers and “ignore the effect of the spinors”. What does that mean? That the natural orbitals are frozen? If this is the case, why? In non-relativistic RDMFT in principle the 2-RDM approximations are functionals of both natural orbitals and occupation numbers. **

We thank the referee for this question. We have realized that it was confusing and we have modified this sentence in the updated manuscript.

**9) It is not completely clear to me if the approximations to the 2-RDM are non relativistic approximations used in a relativistic context or relativistic approximation which are built following similar steps as in the non relativistic case. For example, is one uses only the Coulomb interaction does one retrieve the same expression as in nonrelativistic RDMFT? It doesn’t seem so, since there are the terms L in the total energy. Maybe the authors can clarify this point.
**

We agree with the referee that this point was not made clear enough. Thus, we have included a paragraph explaining that within the np(vp)-ReRDMFT context, the strategy followed to build functional approximations is the same as in the nonrelativistic context. Thus, the functionals presented in this work are a generalization of their nonrelativistic counterparts. Finally, answering the question, the functionals proposed in this work indeed retrieve their nonrelativistic counterparts. In the f(n_i,n_j)-functionals, the L integral vanishes in the nonrelativistic limit upon integration over spin and a purely exchange integral (K_ij) remains active. For PNOFx/GNOF functionals, L integrals also vanish, but the K integral that remains comes from the second term of the third row of Eq. 64. This K integral is usually written as an L integral in the nonrelativistic context to indicate that the two indices of the spinless 2-RDM matrix elements (^2D_ii,jj) refer to an interaction of odd spin type.

10) Related to point 8: could the author estimate the cost of the practical application of the theory?

We would like to thank the referee for this question. We estimate that the cost of the orbital optimization is that of a relativistic MCSCF calculation. The advantage w.r.t. to a MCSCF calculation is that the search for a CI vector is replaced by an optimization of the occupation numbers. We have included this information in the latest version of the manuscript.     **

11) some typos, such as:**
 "settling the theoretical foundation"--->"setting the theoretical foundation"
 "with sets the zero of the energy scale"--->"which sets the zero of the energy scale"
"it possible to distinguish"--->"it is possible to distinguish"
"only up two indices"--->"only up to two indices"
 "a equivalent"---> "an equivalent"

We thank the referee for finding these typos that we have corrected in the updated manuscript.

---

## Round 2 · Referee Report · Anonymous (Referee 4) · 2022-4-29

Report

The authors have addressed all the points I had raised and changed the manuscript accordingly when necessary. As far as I am concerned, the manuscript can be accepted for publication.

---

## Round 2 · Referee Report · Julien Toulouse (Referee 1) · 2022-5-1

Report

The authors have appropriately addressed the points raised by the referees. I recommend publication.
I just have two minor comments below.

Requested changes

- In Eq. (1), there is still a "\alpha_r"
- First line after Eq. (21), there is still a tilde on top of the creation Dirac field operator

---

## Round 2 · Referee Report · Anonymous (Referee 5) · 2022-5-4

Report

The authors have well addressed all issues raised in my previous report.

---

## Editorial Decision

published